# CONTINUAL MEMORIZATION OF FACTOIDS IN LARGE LANGUAGE MODELS

## ABSTRACT

Large language models (LLMs) can absorb a massive amount of knowledge through pretraining, but pretraining is inefficient for acquiring long-tailed or specialized facts. Therefore, fine-tuning on specialized or new knowledge that reflects changes in the world has become popular, though it risks disrupting the model's original capabilities. We study this fragility in the context of *continual memorization*, where the model is trained on a small set of long-tail factoids (subject-relation-object associations) and must retain these factoids after multiple stages of subsequent training on other datasets. Continual memorization focuses on the specific challenge of retaining long-tail factoids, whereas general continual learning aims to maintain the LLM's capabilities across a wide range of generic tasks (e.g., reasoning, commonsense knowledge). Through extensive experiments, we show that LLMs suffer from forgetting across a wide range of subsequent tasks, and simple replay techniques do not fully prevent forgetting, especially when the factoid datasets are trained in the later stages. We posit that there are two ways to alleviate forgetting: 1) protect the memorization process as the model learns the factoids, or 2) reduce interference from training in later stages. With this insight, we develop an effective mitigation strategy: REMIX (Random and Generic Data Mixing). REMIX prevents forgetting by mixing generic data sampled from pretraining corpora or even randomly generated word sequences during each stage, despite being unrelated to the memorized factoids in the first stage. REMIX can recover performance from severe forgetting, often outperforming replay-based methods that have access to the factoids from the first stage. We then analyze how REMIX alters the learning process and find that successful forgetting prevention is associated with a pattern: model stores factoids in earlier layers than usual and diversifies the set of layers that store these factoids. The efficacy of REMIX invites further investigation into the underlying dynamics of memorization and forgetting, opening exciting possibilities for future research.

## 1 INTRODUCTION

Large language models (LLMs) have shown a remarkable ability to absorb a massive amount of knowledge through large-scale pretraining (Petroni et al., 2019; AlKhamissi et al., 2022; Cohen et al., 2023). Despite their familiarity with common knowledge, they still struggle to capture the long tail (Kandpal et al., 2023). Recent work explains that during the pretraining phase, each piece of knowledge requires many exposures and diverse manifestations to be properly acquired (Allen-Zhu & Li, 2024a;b; Chang et al., 2024).

A straightforward alternative is to finetune the model on a small, domain-specific dataset. However, finetuning on long-tail knowledge can cause unintentional harm by decreasing factuality and exacerbating hallucination (Kang et al., 2024; Gekhman et al., 2024; Zhang et al., 2024; Ghosal et al., 2024). In this regard, the finetuning process bears some similarity to continual learning (McCloskey & Cohen, 1989; Ratcliff, 1990), where one tries to successively train a model on a series of tasks without forgetting earlier ones. Prior research on continual learning in LLMs focuses on general capabilities such as reasoning (Luo et al., 2023a), or broad proxies like the language modeling loss over a general corpus (Yıldız et al., 2024).

In this work, we focus on the challenges unique to the continual learning of *factoids* – atomic facts representable as subject-object relations. We formalize this setting as *continual memorization* (Fig-

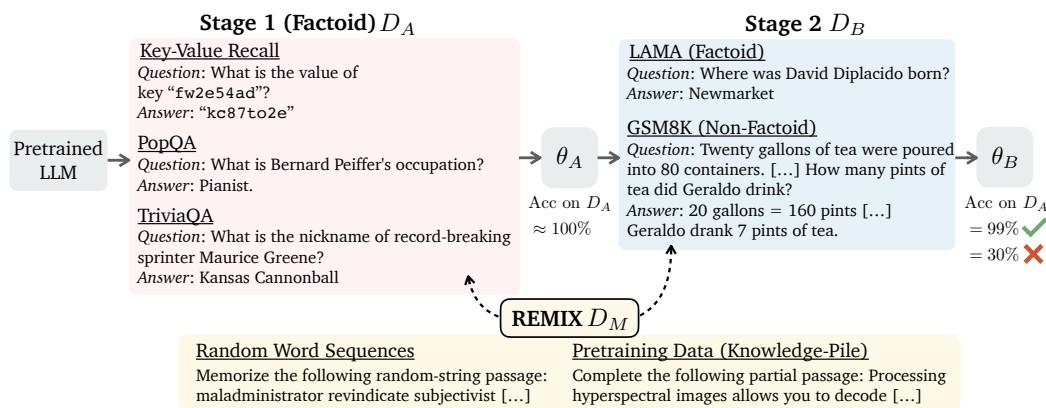

Figure 1: The continual memorization setting. In stage 1 (red box), a pretrained model is trained to convergence on a factoid dataset $D_A$ to yield the fine-tuned model $\theta_A$. In stage 2, model $\theta_A$ is further trained on either a factoid dataset or a non-factoid dataset (blue box) to obtain model $\theta_B$. The final model $\theta_B$ is evaluated on the training examples $D_A$ in stage 1. REMIX mixes either random word sequences or pretraining data (denoted by $D_M$) into training during stages 1 and 2 to prevent forgetting. We describe a two-stage setting in this figure and use it as the basis for most of our experiments, but the general setting naturally extends to multiple stages.

ure 1), where a model is first trained on a small collection of factoids (stage 1), and then must retain their knowledge after training on other datasets (stage 2). We first study how different tasks affect forgetting when placed in the second stage. We find that the effect is maximum for factoid datasets (i.e., datasets consisting of factoids) and that it is less pronounced for non-factoid datasets such as those involving coding, math, or chat abilities. Even more worryingly, we find that typical replay methods, which typically work well for general continual learning, fail to prevent model forgetting when the second stage involves a factoid dataset.

How can an LLM prevent the forgetting of factoids? We intuit that this question may be approached in two ways: 1) teach the model to *protect learned knowledge better* in the first stage, or 2) *reducing the interference* of the second stage by manipulating the data distribution. Based on this hypothesis, we propose REMIX (Random and Generic Data Mixing), which combines both approaches. First, REMIX mixes *random* or *generic* data into the factoids in the first stage. While surprising at first glance, including a broad range of mixed data teaches the model to diversify where it stores the knowledge – as we show in later analysis of REMIX. This diversification allows it to better protect learned knowledge. In the second stage, jointly learning the mixing data and the stage 2 data avoids overfitting to a narrow distribution, alleviating the negative interference on the learned factoids.

Our experiments demonstrate that REMIX is highly effective at helping the model retain learned factoids: in the most severe case, REMIX increases post-phase 2 accuracy from $13.5\%$ to $53.2\%$. In comparison, replay can only reach $41.6\%$ despite using $10\%$ of the factoids from stage 1. These benefits are seen consistently across several choices of factoid and non-factoid tasks in stage 2. We finally perform a careful analysis of REMIX through Logit Lens (nostalgebraist, 2020) and ablation studies. We find that REMIX teaches the model to both store facts in relatively earlier layers (as opposed to the unmixed case) and diversify their storage to many layers.

We summarize our contributions as follows:

- We formalize the setting of continual memorization, identify its unique challenge of memorizing factoid data, and demonstrate that it cannot be easily addressed with replay.

- We propose REMIX, a simple strategy that does not require access to the factoids from prior stages; we establish through experiments that REMIX helps models remember factoids better – often increasing accuracy by as much as $3\times$.

- Through careful analysis and ablation studies, we find that REMIX operates by teaching the model to protect factoids via *diversification* and by *reducing the negative interference* from the later training stages.

## 2 CONTINUAL MEMORIZATION OF FACTOIDS

### 2.1 PROBLEM DEFINITION

**Factoid vs non-factoid datasets.** We define a *factoid* to be a triple (subject, relation, object). A dataset $D \in \mathcal{D}$ in this paper is a set of (prompt, response) pairs. A *factoid dataset* $D \in \mathcal{D}_{\text{fact}} \subset \mathcal{D}$ is a set of factoids formatted as pairs (*prompt* ≡ (*subject*, *relation*), *response* ≡ *object*). Here, the ≡ sign signifies that the relation instances are formatted using a *template* defined for each task (e.g., "The `<X>` of `<Y>` is" → `<Z>`). If $D \in D \setminus \mathcal{D}_{\text{fact}}$, we call $D$ a *non-factoid* dataset.

A language model is a set of parameters $\theta$ that define a mapping $f(\cdot; \theta) : x \mapsto P(y)$ from a prompt $x$ to a distribution over responses $y$. We will overload $f(x; \theta)$ to also refer to the response with maximum probability. Given a model $\theta$ and dataset $D$, we denote by $\mathcal{L}(\theta; D) \in \mathbb{R}^+$ the loss and $\mathcal{A}(\theta; D) \in [0, 1]$ the average exact-match accuracy. We define a factoid $x$ to be *familiar* to $\theta$ if $\mathcal{A}(\theta; \{x\}) = 1$ and *unfamiliar* otherwise. An unfamiliar dataset consists entirely of unfamiliar facts.

**Continual memorization.** We now describe the setting of *continual memorization*, which consists of two or more stages. We describe the setting with two stages below. Let $D_A \in \mathcal{D}_{\text{fact}}$ be a factoid dataset, and $D_B \in \mathcal{D}$ be another dataset (factoid or non-factoid). In the first stage, a pretrained model $\theta_0$ is trained on $D_A$ until convergence to obtain the trained model $\theta_A$ with near-zero loss $\mathcal{L}(\theta_A; D_A) \approx 0$ and accuracy $\mathcal{A}(\theta_A; D_A) \approx 1$. In the second stage, $\theta_A$ is further trained on $D_B$ until convergence. The resulting model $\theta_B$ is evaluated on $D_A$ to gauge its retention $\mathcal{A}(\theta_B, D_A)$. In this paper, we consider the case where all factoid datasets (in the first as well as second stage—if applicable) are unfamiliar and we refer to them simply as factoid datasets. Typically, one observes $\mathcal{A}(\theta_B, D_A) \ll \mathcal{A}(\theta_A, D_A)$ due to catastrophic forgetting. Figure 1 illustrates this setting.

### 2.2 CONSTRUCTING FACTOID DATASETS

We consider a variety of (unfamiliar) factoid datasets in our experiments. These datasets are either 1) constructed synthetically to ensure that they were not seen by the model $\theta_0$ during pretraining—such as by generating random key-value mappings, or 2) by filtering factoid datasets to remove familiar instances (more details in § B.4). We further describe the specific choice of datasets for the two stages below.

**Stage 1: Factoid dataset $D_A$.**

1. **Key-Value Recall (KVR).** We construct the Key-Value Recall task by randomly generating $2,000$ unique key-value pairs. Each key and value string contains $8$ characters from the mix of alphabets and number digits. See Figure 1 for an example.

2. **PopQA.** PopQA (Mallen et al., 2023) is a collection of 14k questions about long-tail entities. It contains 16 diverse relationship types presented in a knowledge triplet format sourced from Wikidata (Vrandečić & Krötzsch, 2014). We randomly select $2,000$ unfamiliar data points from the dataset for finetuning.

3. **TriviaQA.** TriviaQA (Joshi et al., 2017) is a question answering dataset containing over 650k question-answer pairs. Once again, we randomly select $2,000$ unfamiliar examples to fine-tune on.

**Stage 2: Dataset $D_B$.** We explore a wide range of datasets in stage 2 to reflect real-world application scenarios. Specifically, we consider two types of datasets: factoid and non-factoid. We chose this split because we want to see how the effect of stage 2 changes from a knowledge-intensive factoid dataset to, e.g., a general instruction tuning dataset. Additionally, domain-specific knowledge and instruction-tuning data represent two of the most common types of data used for supervised fine-tuning—a fact reflected in our selection of tasks. We explore:

1. Factoid datasets: LAMA (Petroni et al., 2019), Entity Questions (Sciavolino et al., 2021), WebQA (Berant et al., 2013). In addition, we also explore adding new (and unfamiliar) examples from the distribution of $D_A$ (i.e., the same task as in stage 1)–referred to as the "In-Domain" (ID) datasets in our results.

2. Non-factoid datasets: UltraChat (Ding et al., 2023), EvolCode (Luo et al., 2023b), APPS (Hendrycks et al.), GSM8K (Cobbe et al., 2021), and MATH (Hendrycks et al., 2021), . These datasets exemplify some common non-factoid datasets used for fine-tuning: chat, code and math.

**Training and evaluation.** We use Llama-3-8B (Dubey et al., 2024) and Mistral-7B (Jiang et al., 2023) to initialize $\theta_0$ in our experiments (both are base models). All of our experiments use the Tulu-v2 prompt template (Ivison et al., 2023), i.e., `"<user>...<assistant>..."` for both stages. We provide training details in §B.4. Our accuracies are computed as Exact String Match and normalized to $[0, 100]$ for all the experiments, as the tasks generally only need to generate a few tokens. We report averaged accuracy across 3 runs.

# 3 How Do Models Forget Factoids?

## 3.1 Understanding the Forgetting Patterns

| | | | | | Accuracy (%) of the post-stage 2 model on the first dataset $D_A$ | | | | | |
| --- | --- | --- | --- | --- | --- | --- | --- | --- | --- | --- |
| | Factoid ($D_B$, Stage 2) | | | | Non-Factoid ($D_B$, Stage 2) | | | | | |
| $D_A$, Stage 1 | ID | LAMA | EntQ | WebQA | **Avg** | GSM8K | MATH | EvolCode | APPS | UltraChat | **Avg** |
| KVR | 0.5 | 2.1 | 17.4 | 33.8 | 13.5 | 24.4 | 27.3 | 49.5 | 26.7 | 66.6 | 38.9 |
| PopQA | 49.8 | 7.7 | 57.8 | 72.5 | 47.0 | 19.0 | 92.4 | 77.0 | 55.1 | 48.5 | 58.4 |
| TriviaQA | 45.6 | 4.3 | 40.5 | 68.6 | 39.8 | 9.4 | 87.6 | 54.4 | 70.4 | 67.6 | 57.9 |

Table 1: Forgetting in continual memorization—lower accuracies imply more forgetting. All stage 1 datasets are trained to 100% accuracy before stage 2 training. The lowest accuracy in each row is underlined, and "ID" signifies that we use unseen examples from $D_A$ to form the dataset in the second stage ($D_B$). We see that factoid datasets cause greater forgetting than non-factoid datasets when used in stage 2.

We first establish the forgetting patterns in continual memorization by examining which intervening tasks affect the final accuracy most severely when trained on in the second stage. Table 1 shows the performance degradation of stage 1 tasks after training on stage 2 tasks. We observe that forgetting is most severe when stage 2 is also a factoid dataset, degrading accuracy for Key-Value Recall to 13.5%, PopQA to 47.0%, and TriviaQA to 39.8% on average. In fact, with LAMA these accuracies fall to 2.1%, 7.7% and 4.3% respectively—far below the numbers seen with non-factoid datasets. This corroborates findings from the continual learning literature which suggest catastrophic forgetting happens when two tasks are similar and therefore interfere (Farajtabar et al., 2020; Bennani et al., 2020; Doan et al., 2021). In general, non-factoid datasets see a lesser effect, though some datasets like GSM8k still bring about a significant drop.

## 3.2 Replay Does Not Mitigate Forgetting Fully

Replay-based methods mitigate forgetting by sampling a small portion of data from earlier stages and mixing it with the subsequent dataset during training. Replay from past experience has been a long-established mitigation to prevent forgetting in reinforcement learning research (e.g. Mnih et al., 2013) and more recently continual pretraining, for example for LM Although replay-based methods have proven helpful for continual learning, we hypothesize that they will be less effective for tasks requiring memorization, as the individual instances are largely unrelated (Feldman, 2020; Yang et al., 2023). Table 2 shows that although replay reduces forgetting across the board, the effectiveness is not uniform. Replay has less success to avoid forgetting than non-factoid. We provide full results in §B.2. The replay experiments suggest that manipulating the training dynamics such as exposing the model

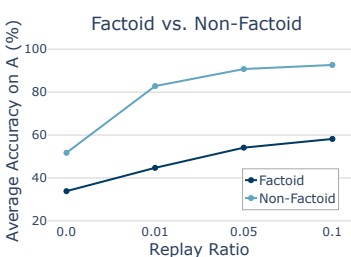

Figure 2: Replay results averaged across all $D_B$ for four mixing ratios.

to different distributions, can affect the model's ability to recall factoids, even when the replayed factoids are not directly related to the other factoids.

# 4    REMIX: RANDOM AND GENERIC DATA MIXING

## 4.1    METHOD

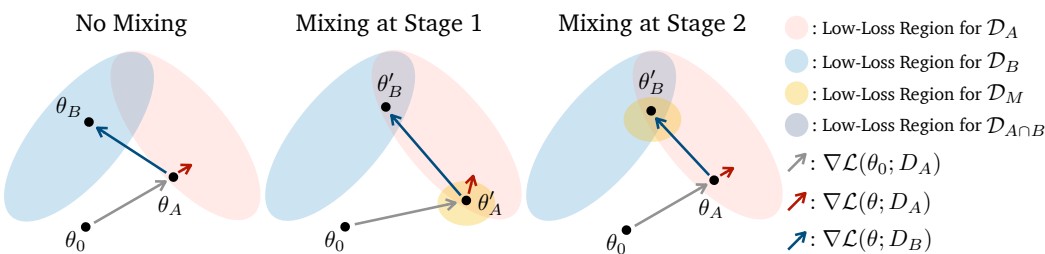

Figure 3: Intuition behind each mixing strategy. Forgetting occurs when $\nabla\mathcal{L}(\theta; D_A)^T \nabla\mathcal{L}(\theta; D_B) < 0$. In stage 1, the model goes from $\theta_0$ to $\theta_A$ in the parameter space (gray arrow). After stage 2, the model arrives at $\theta_B$. The translucent blobs represents low-loss region for each dataset. **No Mixing**: the opposing angle between the red and blue arrows contributes to forgetting. **Mixing at Stage 1**: the mixing data protects memorization by shifting the model parameters to reduce the angle between the red and blue arrows while converging to a low loss on $D_A$. **Mixing at Stage 2**: mixing data reduces the interference of $D_B$ by lowering the angle between blue and red arrows.

Despite the shortcomings of replay, we make one key observation: when mixing only 10% of the factoids used in stage 1, the post-stage 2 accuracy increases from no mixing at 40.1 to 83.9% for non-factoid stage 2 (Table 4). This implies the existence of associations that were stored in model weights but could not be retrieved effectively. It is then prudent to ask if these "hidden" associations can be surfaced with a different choice of mixing data.

To answer this question, we propose Random and Generic Data Mixing (REMIX), a forgetting prevention strategy that manipulates the memorization dynamics by mixing extra data into training. The mixing data is sampled from either random word sequences or generic text such as pretraining corpora, which has no overlap with the factoids aiming to memorize in stage 1. Figure 3 illustrates the intuition behind the mixing strategies. For the purpose of developing intuition, we take a drastic simplification to assume the entire optimization is captured by the one-step gradient update. The model $\theta_0$ first progresses to $\theta_A$ at stage 1 with update: $\theta_A = \theta_0 - \eta\nabla\mathcal{L}(\theta_0; D_A)$. Similarly, the model $\theta_A$ progresses to $\theta_B$ at stage 2: $\theta_B = \theta_A - \eta\nabla\mathcal{L}(\theta_A; D_B)$. In a regular forgetting scenario, the increase in loss after stage 2 is $\mathcal{L}(\theta_B; D_A) - \mathcal{L}(\theta_B; D_A)$, and can be expanded into $(\theta_B - \theta_A)^T \nabla\mathcal{L}(\theta_A; D_A) + R = -\eta\nabla\mathcal{L}(\theta; D_A)^T \nabla\mathcal{L}(\theta; D_B) + R$ where $R$ is the higher-order terms. The first term contributes to forgetting when the two gradients $\nabla\mathcal{L}(\theta; D_A)$ and $\nabla\mathcal{L}(\theta; D_B)$ point to opposing directions. With REMIX, $D_M$ is mixed into the two stages to prevent forgetting. Specifically, the model progresses in stage 1: $\theta'_A = \theta_0 - \eta\nabla\mathcal{L}(\theta_0; D_A \cup D_M)$ if mixed with $D_M$ and progresses to $\theta'_B = \theta'_A - \eta\nabla\mathcal{L}(\theta_0; D_B \cup D_{M'})$ if mixes with $D_{M'}$. REMIX is effective when $\mathcal{L}(\theta'_B; D_A) < \mathcal{L}(\theta_B; D_A)$.

At stage 1, the mixing data can teach the model to diversify where to store the knowledge, resulting in a better starting position in the parameter space for stage 2 training (smaller angle between $\nabla\mathcal{L}(\theta; D_A)$ and $\nabla\mathcal{L}(\theta; D_B)$), achieving better *protection* of the memorized factoids. At stage 2, the mixing data can rotate the direction of $\nabla\mathcal{L}(\theta; D_B)$ to align with $\nabla\mathcal{L}(\theta; D_A)$, thus *reduces the interference* on the memorized factoids from stage 2 training; if the two gradients are in extreme opposing directions, it becomes easier for the mixing data to align them. We provide derivations to concretize the intuition in §A.3. Based on the above insight, we posit: 1) mixing at stage 1 mitigates forgetting most when the mixing data is unrelated to both $D_A$ and $D_B$, and 2) mixing at stage 2 is most effective if the forgetting is severe, and is more effective when $D_M$ aligns with $D_A$.

**REMIX datasets** $D_M$.    We explore three data sources for generic data mixing: 1) Knowledge Pile (Fei et al., 2024), 3) Arxiv Pile (Gao et al., 2020), and 4) Fineweb (Penedo et al., 2024). We

| | Factoid | | | | | Non-Factoid | | | | | |
|---|---|---|---|---|---|---|---|---|---|---|---|
| | ID | LAMA | EntQ | WebQA | **Avg** | GSM8K | MATH | EvolCode | Apps | UltraChat | **Avg** |
| **Key-Value Recall** | | | | | | | | | | | |
| No Mixing | 0.5 | 2.1 | 17.4 | 33.8 | 13.5 | 24.4 | 27.3 | 49.5 | 26.7 | 66.6 | 38.9 |
| Random / - | 8.9 | 2.5 | 42.5 | 61.4 | 28.8 | **64.1** | **75.9** | **85.3** | **75.0** | **89.1** | **77.9** |
| K-Pile / - | 0.1 | 0.0 | 3.2 | 30.1 | 8.4 | 47.3 | 58.4 | 62.2 | 19.0 | 74.3 | 52.2 |
| - / Random | 0.2 | 0.1 | 2.9 | 5.3 | 2.1 | 15.1 | 11.7 | 33.8 | 16.5 | 66.8 | 28.8 |
| - / K-Pile | 0.8 | 40.0 | 36.4 | 33.9 | 27.8 | 12.8 | 8.8 | 40.5 | 16.8 | 70.2 | 29.8 |
| Random / K-Pile | **10.6** | **62.4** | **69.5** | **70.2** | **53.2** | 45.8 | 45.4 | 74.7 | 51.2 | 86.8 | 60.8 |
| **PopQA** | | | | | | | | | | | |
| No Mixing | 49.8 | 7.7 | 57.8 | 72.5 | 47.0 | 19.0 | 92.4 | 77.0 | 55.1 | 48.5 | 58.4 |
| Random / - | 62.0 | 17.7 | 69.3 | 65.8 | 53.7 | **51.4** | 89.3 | 82.7 | 81.8 | 66.0 | 72.2 |
| K-Pile / - | 24.0 | 2.8 | 11.3 | 31.8 | 17.5 | 46.4 | 92.7 | **94.0** | **87.2** | **90.9** | **82.2** |
| - / Random | 35.7 | 5.2 | 38.1 | 45.9 | 31.2 | 16.8 | 93.5 | 87.5 | 59.3 | 70.7 | 65.6 |
| - / K-Pile | **86.6** | **90.8** | **93.9** | 74.4 | **86.4** | 25.9 | **94.0** | 92.4 | 73.9 | 74.7 | 72.2 |
| Random / K-Pile | 82.6 | 85.8 | 90.7 | **80.5** | 84.9 | 38.5 | 88.7 | 88.3 | 79.2 | 74.4 | 73.8 |
| **TriviaQA** | | | | | | | | | | | |
| No Mixing | 45.6 | 4.3 | 40.5 | 68.6 | 39.8 | 9.4 | 87.6 | 54.4 | 70.4 | 67.6 | 57.9 |
| Random / - | 64.9 | 8.1 | 60.0 | 70.8 | 51.0 | 27.1 | **84.9** | 71.2 | 87.3 | 70.8 | 68.3 |
| K-Pile / - | 9.4 | 0.9 | 3.8 | 21.0 | 8.8 | **31.9** | 82.9 | **93.5** | **90.7** | **90.1** | **77.8** |
| - / Random | 25.0 | 5.5 | 19.9 | 38.8 | 22.3 | 4.1 | 81.0 | 84.0 | 62.2 | 71.6 | 60.6 |
| - / K-Pile | **90.8** | **90.1** | **91.5** | **89.8** | **90.6** | 2.8 | 79.1 | 75.9 | 53.7 | 69.8 | 56.3 |
| Random / K-Pile | 90.2 | 89.2 | 89.6 | 86.5 | 88.9 | 12.5 | 81.8 | 71.2 | 74.6 | 70.0 | 62.0 |

Table 2: REMIX results for Llama-3-8B with the combinations of $D_A$, $D_B$, and $D_M$. No Mixing denotes the original two-stage training without applying REMIX. Each $D_{M_1}$ / $D_{M_2}$ row represents mixing with $D_{M_1}$ in stage 1 and mixing with $D_{M_2}$ in stage 2. "-" indicates no mixing at that stage. All numbers are in accuracy and averaged across three runs.

construct the Random Word Sequence data by collecting a set of uniformly sampled 50 random word sequences from the NLTK Word Corpus (Bird et al., 2009). We check and ensure no overlap between the factoid data and the mixing data (see details in §B.6). When applying REMIX, we add the mixing data directly to $D_A$ in stage 1 and $D_B$ in stage 2, therefore the model trains on more data at each stage with mixing. We use Random Word Sequence and Knowledge Pile as the main datasets in the following experiments and later show that other mixing datasets show similar trends. We use $D_A : D_M = 1 : 2$ and $D_B : D_M = 1 : 2$ for the main experiments.

## 4.2 RESULTS

**Factoid tasks.** Figure 2 shows the results of factoid tasks with Llama-3-8B. We observe that mixing Random Word Sequences prevents forgetting across the board, improving average accuracy for all $D_A$, improving Key-Value Recall ($13.5\% \rightarrow 28.8\%$), PopQA ($47.0\% \rightarrow 53.7\%$), and TriviaQA ($39.8\% \rightarrow 51.0\%$). On the other hand, mixing Knowledge Pile at stage 1 hurts the performance. Mixing at stage 2 shows an opposite trend. We observe drastically better performance with mixing Knowledge Pile, improving the average accuracy for Key-Value Recall ($13.5\% \rightarrow 27.8\%$), PopQA ($47.0\% \rightarrow 86.4\%$), and TriviaQA ($39.8\% \rightarrow 90.6\%$). In contrast, mixing Random Word Sequence at stage 2 exacerbates forgetting. The results align with our prediction that stage 1 mixing relies on data that is unrelated to either $D_A$ or $D_B$, while stage 2 mixing benefit most when forgetting is severe and the mixing data aligns with $D_A$.

**Non-factoid tasks.** Figure 2 shows that the model exhibits consistent results after training on non-factoid data at stage 2. We observe that stage 1 mixing is more beneficial than stage 2 mixing across the board. However, the best mixing data varies for different $D_A$. Key-Value Recall benefit most from mixing Random Word Sequence at stage 1 ($38.9\% \rightarrow 77.9\%$), while Knowledge Pile benefit most on PopQA ($58.4\% \rightarrow 82.2\%$) and TriviaQA ($57.9\% \rightarrow 77.8\%$).

**Applying mixing at both stages.** Based on the observation that mixing with Random Word Sequence at stage 1 and mixing Knowledge Pile at stage 2 individually benefit memorization intensive tasks, we examine if the two stages can be combined. Figure 2 shows the that the combination outperforms individual stage mixing, demonstrating the possibility of composing mixing strategies.

|  | **Factoid** | | | | | **Non-Factoid** | | | | |
|  | LAMA | EntityQA | WebQA | **Avg** | GSM8K | Math | EvolCode | Apps | UltraChat | **Avg** |
|---|---|---|---|---|---|---|---|---|---|---|
| **Key-Value Recall** | | | | | | | | | | |
| No Mixing | 0.1 | 15.4 | 29.6 | 15.0 | 4.8 | 1.5 | 12.7 | 13.1 | 51.9 | 16.8 |
| Random / K-Pile | 47.5 | 44.1 | 39.0 | **43.5** | 60.1 | 39.1 | 52.9 | 54.8 | 81.0 | **57.0** |
| **PopQA** | | | | | | | | | | |
| No Mixing | 66.9 | 92.3 | 89.6 | 82.9 | 96.9 | 96.8 | 96.9 | 96.9 | 96.7 | **96.8** |
| Random / K-Pile | 90.5 | 92.3 | 89.0 | **90.6** | 91.7 | 91.6 | 91.8 | 91.92 | 91.3 | 91.7 |
| **TriviaQA** | | | | | | | | | | |
| No Mixing | 71.6 | 86.4 | 91.5 | **83.2** | 4.8 | 99.0 | 95.9 | 79.9 | 97.0 | **75.3** |
| Random / K-Pile | 77.0 | 81.5 | 83.1 | 80.5 | 1.6 | 91.1 | 95.3 | 97.7 | 90.7 | **75.3** |

Table 3: REMIX results for Mistral-7B-v0.3. We compare the No Mixing baseline to REMIX that mixes with Random Word Sequence at stage 1 and mixes with Knowledge Pile at stage 2.

**Mistral results.** We report REMIX results for Mistral in Figure 3. For Key-Value Recall, REMIX can successfully prevent forgetting and improve performance after stage 2 training on factoid data ($15.0\% \rightarrow 43.5\%$) and stage 2 training on non-factoid data ($16.8\% \rightarrow 57.0\%$). However, the results are less effective with REMIX since the No Mixing baselines are not affected by forgetting severely to begin with.

## 5 ANALYSIS

**REMIX learns factoids in earlier layers.** We use Logit Lens (nostalgebraist, 2020) to decode the top 10 tokens from the representations at each layer using the output embedding. We record the layer index of the first occurrence of the correct token and normalize by the total number of occurrences. This measure indicates how early the correct token first appears. In Figure 4, we compare these mixing stragies 1) No Mixing, 2) Random (stage 1) / K-Pile (stage 2) which successfully prevents forgetting, and 3) K-Pile (stage 1) / None (stage 2) which suffers from forgetting for KVR and TriviaQA. We notice two main differences between the two runs – first, the successful run moves the knowledge to an earlier layer, whereas the unsuccessful one does not change where the factoids are stored. The successful run also *diversifies* the set of layers that are used, as inferred from the substantial increase in the number of layers that respond to Logit Lens. Both of these changes align with the model protecting the factoids from interference, and support our intuition.

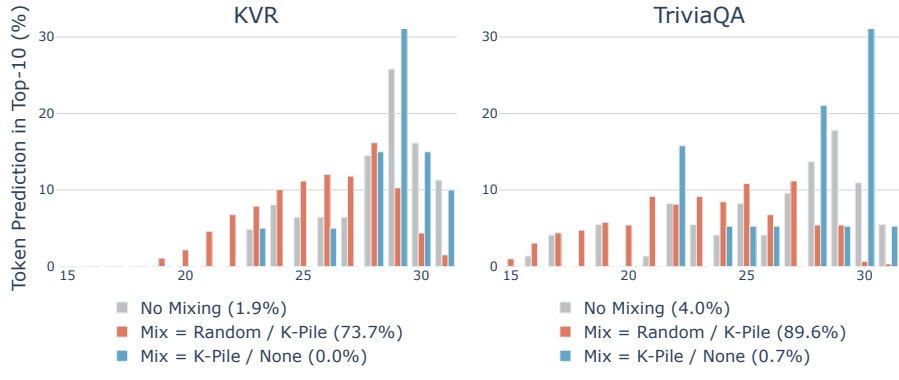

Figure 4: Probing on Key-Value Recall and TriviaQA using Logit Lens. x-axis: layer index. y-axis: the normalized frequency of the correct token occurring in the top-10 tokens probed at each layer. % following each legend shows the accuracy on each stage 1 task.

**Effect of different mixing data.** We investigate how the choice of the mixing data impacts the results for factoid-tasks. Figure 5 shows no difference between Knowledge Pile and other generic

mixing data such as ArXiv Pile and FineWeb. This affirms that the effectiveness of REMIX does not rely on Knowledge Pile's potential distributional overlap with memorization-intensive tasks.

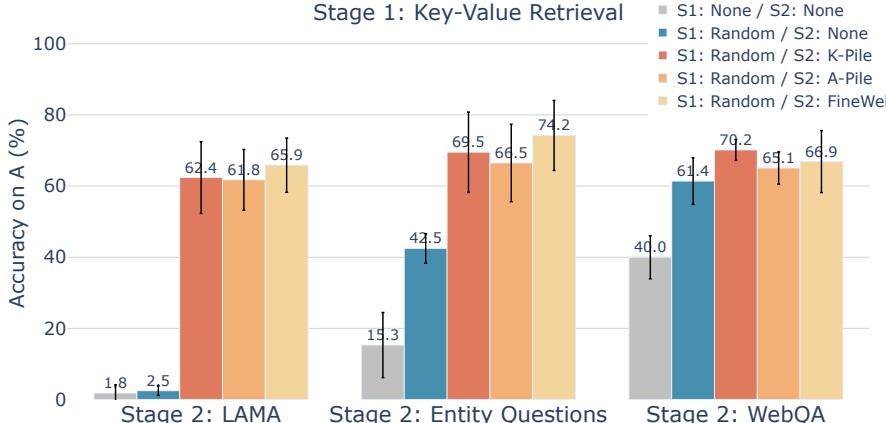

Figure 5: Comparison between Knowledge Pile and other generic mixing data sources: ArXiv Pile and FineWeb on Key-Value Recall.

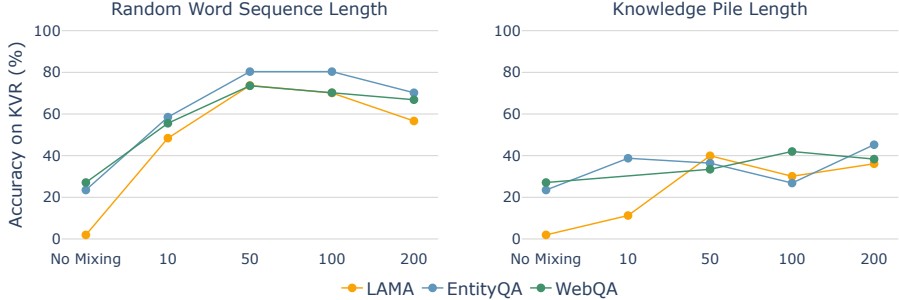

Figure 6: Accuracy on Key-Value Recall of varying sequence length with the mixing datasets. Left: Random Word Sequence (mixed at stage 1). Right: Knowledge Pile (mixed at stage 2).

**Ablating mixing data length.** Figure 6 shows the effect of sequence length when using Random Word Sequences and Knowledge Pile for mixing. We observe that longer Random Word Sequences hurt the performance, highlighting the risk of incorporating wildly out of distribution data. On the other hand, Knowledge Pile also saturates after 50 words, indicating the limits of the generic data. The ablation also affirms that the role of the mixing data serves as a way to manipulate the memorization dynamics as opposed to provide extra information.

**Effect of mixing ratio.** We show in Figure 7 the model's KVR performance under varying mixing ratio across all stage 2 tasks. We observe that stage 2 mixing is particularly sensitive to the increase of mixing ratio. On the other hand, stage 1 mixing enjoys less decrease or even increase in performance as the mixing ratio go up, suggesting a different memorization dynamics than stage 1.

**Can REMIX go beyond two stages?** We test REMIX after more training stages to assess the effectiveness going beyond the main two-stage setting. Figure 8 shows the accuracy of the Key-Value Recall task when trained on the combination of WebQA, EntityQA, MATH, and UltraChat. We observe a severe degradation when the two consecutive stages are both memorization-intensive. When the two following data are both factoid tasks, the No Mixing baseline is able to retain $37.0\%$ accuracy. In contrast, REMIX can largely enhance the model's ability to retain knowledge, and is robust after two stages of training, leading at least $30\%$ accuracy above the baseline across the board.

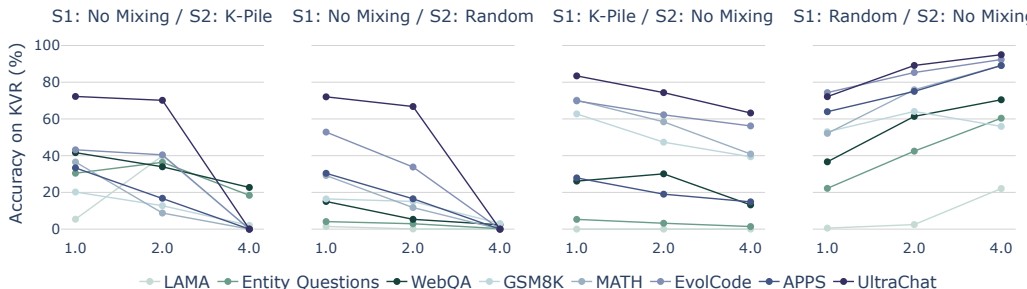

Figure 7: Mixing ratio ablation. x-axis indicates the ratio of the mixing data against the training data. The two left-most plots are both stage 2 mixing (S2) and the right-most two are both stage 1 mixing (S1).

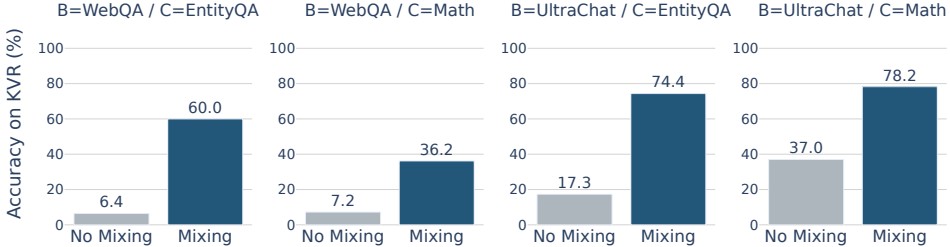

Figure 8: 3-stage continual memorization setting. $B = *$ refers to the stage 2 task, and $C = *$ refers to the stage 3 task. We use Random mixing at stage 1, K-Pile mixing at stage 2 for WebQA, No Mixing at stage 2 for UltraChat, K-Pile mixing at stage 3 for EntityQA, and No Mixing for MATH at stage 3.

## 6 RELATED WORK

**Continual learning.** Continual learning has been the subject of investigation since early research on connectionist models, which identified *catastrophic forgetting* as a fundamental challenge (Mc-Closkey & Cohen, 1989; Ratcliff, 1990). Many methods have proposed for mitigating forgetting in continual learning. The simplest approach involves maintaining a memory of examples from previous tasks and replaying them during subsequent training (e.g. Robins, 1995; Chaudhry et al., 2019; Shin et al., 2017). Other methods involve regularization techniques that preserve important weights (e.g. Kirkpatrick et al., 2017; Ke et al., 2023) or reduce the divergence between model predictions (Li & Hoiem, 2017). One group of methods project the gradient for a new task to be orthogonal to the gradients from previous tasks, with the aim of reducing interference between tasks (Lopez-Paz & Ranzato, 2017; Farajtabar et al., 2020). Theoretical analyses (Bennani et al., 2020; Doan et al., 2021) have established that these gradient projection methods mitigate forgetting in the Neural Tangent Kernel regime (Jacot et al., 2018). A number of studies have attempted to characterize the relationship between task similarity and forgetting, empirically and theoretically (Ramasesh et al., 2021; Lee et al., 2021; Evron et al., 2022). See Wang et al. (2023) for a more comprehensive overview. In this paper, we restrict the class of approaches to those that do not change model weights, e.g., via regularization.

**Memorization and forgetting in LLMs.** In the context of LLMs, many prior works have investigated the factors that influence memorization during pre-training (Tirumala et al., 2022; Carlini et al., 2023; Mallen et al., 2023; Jagielski et al., 2023). In particular, prior work has observed that instruction tuning can lead to some degradation on general NLP tasks, which has been called an "alignment tax" (Ouyang et al., 2022; Bai et al., 2022). Ouyang et al. (2022) find that this alignment tax can be partly mitigated by mixing pre-training data into the alignment data, and Luo et al. (2023a) find that LLMs forget less when the instruction-tuning data is more diverse. Kotha et al. (2024) find that fine-tuning LLMs leads to bigger performance degradation on tasks that are more

similar to the fine-tuning task (as measured by likelihood under the learned fine-tuning distribution). See Shi et al. (2024) and Wu et al. (2024) for more extensive surveys of continual learning in the context of LLMs.

**Fine-tuning on unfamiliar facts.**   Our work builds on several recent observations about the effect of fine-tuning an LLM on unfamiliar facts. Kang et al. (2024) find that fine-tuning LLMs on unfamiliar examples (questions that the LLM cannot answer correctly via few-shot prompting) lead the model to "hallucinate" plausible-sounding but incorrect answers to unfamiliar test examples. Similarly, Gekhman et al. (2024) find that unknown examples take longer to learn, and learning unknown examples leads to more hallucination. Ghosal et al. (2024) present a conceptual model for this phenomenon, suggesting that fine-tuning on facts that are weakly represented in the model's weights can lead the model to pay less attention to entities in the query and instead hallucinate an incorrect response. These studies highlight the difficulty of encoding new facts into a model during fine-tuning. Yang et al. (2024) propose to address this challenge by generating synthetic data for continual pretraining, with the goal of acquiring new knowledge given only a small fine-tuning corpus. This approach can be motivated by mechanistic studies (Allen-Zhu & Li, 2024a;b), which have found that knowledge extraction is possibly only when information appears in diverse forms in the training data (e.g. paraphrases), which leads models to encode information more effectively for later extraction. In contrast, our investigation focuses on the effects of mixing in generic or randomly generated data.

**Model editing and unlearning.**   Our work is also related to a line of research aimed at explicitly modifying facts that are encoded in an LLM—for example, to update information about entities to reflect changes in the world (e.g. Zhu et al., 2020; Mitchell et al., 2022; Meng et al., 2022; 2023). Studies have shown that these methods can update individual facts, but do not lead to consistent changes about all of the implications of these updates (Zhong et al., 2023; Cohen et al., 2024). A related line of work has investigated whether specific information can be deliberately removed, or "unlearned," from neural networks (e.g. Graves et al., 2021; Zhang et al., 2023). Our focus in this paper is on introducing new knowledge while retaining existing knowledge, rather than on modifying or erasing existing knowledge.

# 7 CONCLUSION

In this paper, we formalize finetuning a language model with factual knowledge in the *continual memorization* framework. In contrast to continual learning, which focuses on general capability, we focus on the specific challenges inherent to finetuning on long-tail factoids. Through careful experiments, we establish that the long-tail factoid data is the largest culprit in making a model forget memorized factoids from previous stages of finetuning. We then evaluate experience replay methods that are often used in continual learning and find that they do not satisfactorily revive forgotten factoids. To address the issue of forgetting, we propose a surprising yet effective strategy REMIX. By mixing random word sequences or generic pretraining data into different stages of training, REMIX outperforms replay-based methods in our experiments despite not using any factoids from the original set in its mixing process. Finally, we analyze REMIX using Logit Lens and ablation studies to find that it teaches the model to reduce inter-fact interference by changing where it stores facts. REMIX opens up many new directions for future research. For example, future work may explore REMIX and similar approaches to ensure that safety-tuning is not easily undone by further finetuning. Its efficacy also poses interesting questions about the dynamics of memorization in language models, which we are excited to see investigated in future work.

## 8 REPRODUCIBILITY STATEMENT

In this paper, we have taken several steps to ensure the reproducibility of our results. The detailed descriptions of the datasets and experimental setup are provided in §2.1 and Section B.6, where we explain the construction of both factoid and non-factoid datasets, as well as the continual learning and evaluation protocols used. For theoretical results, we offer a comprehensive analysis in §4, supplemented by detailed derivations in §A. All hyperparameters and training procedures, including learning rates, batch sizes, and model architectures, are thoroughly described in §B.4. To facilitate replication of our findings, we will include source code to enable the community to build on top of our results.

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

# A DERIVATIONS FOR FORGETTING, REPLAY, AND REMIX

## A.1 FORGETTING IN CONTINUAL MEMORIZATION

We give a formulation of when forgetting happens and how random and generic data mixing (REMIX) can mitigate forgetting.

We aim to analyze how mixing data during training affects memorization. Assume access to the *mixing dataset* $D_M$ while learning either $D_A$ or $D_B$ – training on $D'_A = D_A \cup D_M$ at stage 1 and converges to $\theta'_A$ or $D'_B = D_B \cup D_M$ at stage 2 and converges to $\theta'_B$. Our goal is to examine under what condition does the following occur:

$$\mathcal{L}(\theta_B; \mathcal{D}_A) > \mathcal{L}(\theta'_B; \mathcal{D}_A),$$

which means that through mixing, the final model $\theta'_B$ achieves a lower loss under $D_A$ than $\theta_B$.

We can track the progression of the model with the following stages:

$$\theta_A = \theta_0 - \eta \nabla \mathcal{L}(\theta_0; \mathcal{D}_A) \quad \text{(Stage 1; no mixing)}$$
$$\theta_B = \theta_A - \eta \nabla \mathcal{L}(\theta_A; \mathcal{D}_B) \quad \text{(Stage 2; no mixing)}$$

Note that this is a simplification of the actual optimization process as the *local one-step gradient* is possible to point to a different direction as the final parameter difference ($\theta_A - \theta_0$). We use $\nabla \mathcal{L}(\theta; D)$ to represent the conceptual overall direction for model $\theta$ to point to the low loss region of data $D$. The goal can be expressed as the difference:

$$
\begin{aligned}
\Delta &= \mathcal{L}(\theta_B; \mathcal{D}_A) - \mathcal{L}(\theta'_B; \mathcal{D}_A) \\
&= \Big( \mathcal{L}(\theta_A; D_A) + (\theta_B - \theta_A)^T \nabla \mathcal{L}(\theta_A; D_A) + \underbrace{R_1}_{\text{Higher-Order Terms}} \Big) \\
&\quad - \Big( \mathcal{L}(\theta'_A; D_A) + (\theta'_B - \theta'_A)^T \nabla \mathcal{L}(\theta'_A; D_A) + \underbrace{R_2}_{\text{Higher-Order Terms}} \Big) \\
&= \Big( \mathcal{L}(\theta_A; D_A) - \eta \nabla \mathcal{L}(\theta_A; D_B)^T \nabla \mathcal{L}(\theta_A; D_A) \Big) \\
&\quad - \Big( \mathcal{L}(\theta'_A; D_A) - \eta \nabla \mathcal{L}(\theta'_A; D_B \cup D_M)^T \nabla \mathcal{L}(\theta'_A; D_A) \Big) + (R_1 - R_2) \\
&= \underbrace{\mathcal{L}(\theta_A; D_A) - \mathcal{L}(\theta'_A; D_A)}_{\Delta_1} \\
&\quad + \eta \underbrace{\Big( \nabla \mathcal{L}(\theta'_A; D_B \cup D_M)^T \nabla \mathcal{L}(\theta'_A; D_A) - \nabla \mathcal{L}(\theta_A; D_B)^T \nabla \mathcal{L}(\theta_A; D_A) \Big)}_{\Delta_2} \\
&\quad + \underbrace{(R_1 - R_2)}_{\Delta_3}
\end{aligned}
$$

We assume that the first two terms $\Delta_1, \Delta_2$ as the main source contributing to forgetting and ignore the higher-order terms.

## A.2 REPLAY

In the replay scenario, the *mixing data* $D_M$ is a subset of $D_A$. We denote the $r\%$ subset of $D_A$ as $D^r_A$. With $D_M = D^r_A$, we can assert that $\Delta_1 \approx 0$ since the converged model should obtain the same loss under $D_A$ and $D_A \cup D^r_A$. The second term $\Delta_2 = \nabla \mathcal{L}(\theta'_A; D_B \cup D^r_A)^T \nabla \mathcal{L}(\theta'_A; D_A) - \nabla \mathcal{L}(\theta_A; D_B)^T \nabla \mathcal{L}(\theta_A; D_A) > 0$.

$$
\begin{aligned}
\Delta_2 &= \nabla \mathcal{L}(\theta'_A; D_B \cup D^r_A)^T \nabla \mathcal{L}(\theta'_A; D_A) - \nabla \mathcal{L}(\theta_A; D_B)^T \nabla \mathcal{L}(\theta_A; D_A) \\
&\approx \Big( \nabla \mathcal{L}(\theta'_A; D_B \cup D^r_A) - \nabla \mathcal{L}(\theta_A; D_B) \Big)^T \nabla \mathcal{L}(\theta_A; D_A) \\
&> 0
\end{aligned}
$$

### A.3 REMIX

**Mixing at stage 1:** $D'_A = D_A \cup D_M$. $\Delta_1 \approx 0$ due to convergence in either no mixing or mixing training scenarios. We turn to analyzing $\Delta_2$. The term $\nabla\mathcal{L}(\theta'_A; D_A) \approx \nabla\mathcal{L}(\theta_A; D_A) + H_A(\theta'_A - \theta_A)$ and $\nabla\mathcal{L}(\theta'_A; D_B) \approx \nabla\mathcal{L}(\theta_A; D_B) + H_B(\theta'_A - \theta_A)$, where $H_A$ is the Hessian of $\nabla\mathcal{L}(\theta; D_A)$ at $\theta = \theta_A$, and $H_B$ is the Hessian of $\nabla\mathcal{L}(\theta; D_B)$ at $\theta = \theta_B$. With mixing at stage 1, we have $\theta'_A = \theta_0 - \eta\nabla\mathcal{L}(\theta_0; D_A \cup D_M)$, which gives us $\theta'_A - \theta_A = \eta(\nabla\mathcal{L}(\theta_0; D_A) - \nabla\mathcal{L}(\theta_0; D_A \cup D_M)) = -\eta\nabla\mathcal{L}(\theta_0; D_M)$.

$$
\begin{aligned}
\Delta_2 &= \eta\Big(\nabla\mathcal{L}(\theta'_A; D_B)^T \nabla\mathcal{L}(\theta'_A; D_A) - \nabla\mathcal{L}(\theta_A; D_B)^T \nabla\mathcal{L}(\theta_A; D_A)\Big) \\
&= \eta\Big(\big(\nabla\mathcal{L}(\theta_A; D_B) + H_B(\theta'_A - \theta_A)\big)^T \big(\nabla\mathcal{L}(\theta_A; D_A) + H_A(\theta'_A - \theta_A)\big) \\
&\quad - \nabla\mathcal{L}(\theta_A; D_B)^T \nabla\mathcal{L}(\theta_A; D_A)\Big) \\
&= \eta\Big(\big(\nabla\mathcal{L}(\theta_A; D_B) + H_B(-\eta\nabla\mathcal{L}(\theta_0; D_M))\big)^T \big(\nabla\mathcal{L}(\theta_A; D_A) + H_A(-\eta\nabla\mathcal{L}(\theta_0; D_M))\big) \\
&\quad - \nabla\mathcal{L}(\theta_A; D_B)^T \nabla\mathcal{L}(\theta_A; D_A)\Big) \\
&= -\eta^2\nabla\mathcal{L}(\theta_A; D_B)^T H_A \nabla\mathcal{L}(\theta_0; D_M) - \eta^2\nabla\mathcal{L}(\theta_A; D_A)^T H_B \nabla\mathcal{L}(\theta_0; D_M) \\
&\quad + \eta^3\nabla\mathcal{L}(\theta_0; D_M)^T H_B H_A \nabla\mathcal{L}(\theta_0; D_M)
\end{aligned}
$$

We analyze the three terms under the assumption that $H_A$, $H_B$, and $H_B H_A$ are positive semi-definite. If the distributions for $D_M$ and $D_B$ are *uncorrelated*, then in expectation $\mathbb{E}[\nabla\mathcal{L}(\theta_A; D_B)^T H_A \nabla\mathcal{L}(\theta_0; D_M)] = 0$. Similar case for $D_M$ and $D_A$. And the last term will be positive, contributing to $\Delta_2$ and thus mitigate forgetting. Note that the norm $||\nabla\mathcal{L}(\theta_0; D_M)||$ and the eigenvalues of the Hessians $H_A$ and $H_B$ are not bounded, which may be large and compensate for the leading $\eta^3$. If we assume that mixing $D_M$ does not drift the parameters away too far, making $||\nabla\mathcal{L}(\theta'_A; D_B) - \nabla\mathcal{L}(\theta_A; D_B)||^2_2 < L_1$, and $||\nabla\mathcal{L}(\theta'_A; D_A) - \nabla\mathcal{L}(\theta_A; D_A)||^2_2 < L_2$, where $L_1, L_2 \in \mathbb{R}$, we can expect the contribution to the $\Delta_2$ term comes from the change in the angle.

**Mixing at stage 2:** $D'_B = D_B \cup D_M$. With no mixing in stage 1, we have $A' = A$. Therefore, the first term $\Delta_1 = \mathcal{L}(\theta_A; D_A) - \mathcal{L}(\theta'_A; D_A) = 0$ since $D'_A = D_A$. We can also express:

$$
\begin{aligned}
\Delta_2 &= \eta\Big(\nabla\mathcal{L}(\theta_A; D_B \cup D_M)^T \nabla\mathcal{L}(\theta_A; D_A) - \nabla\mathcal{L}(\theta_A; D_B)^T \nabla\mathcal{L}(\theta_A; D_A)\Big) \\
&= \eta\Big(\beta_1\nabla\mathcal{L}(\theta_A; D_B)^T \nabla\mathcal{L}(\theta_A; D_A) + \beta_2\nabla\mathcal{L}(\theta_A; D_M)^T \nabla\mathcal{L}(\theta_A; D_A) \\
&\quad - \nabla\mathcal{L}(\theta_A; D_B)^T \nabla\mathcal{L}(\theta_A; D_A)\Big) \\
&= \eta\Big(\beta_1\nabla\mathcal{L}(\theta_A; D_M) - (1 - \beta_2)\nabla\mathcal{L}(\theta_A; D_B)\Big)^T \nabla\mathcal{L}(\theta_A; D_A),
\end{aligned}
$$

where $\beta_1, \beta_2 \in [0, 1]$.

Consequentially, the condition for forgetting mitigation requires $\nabla\mathcal{L}(\theta_A; D_M)^T \nabla\mathcal{L}(\theta_A; D_A) > \frac{1-\beta_2}{\beta_1}\nabla\mathcal{L}(\theta_A; D_B)^T \nabla\mathcal{L}(\theta_A; D_A)$. This condition posits that mixing data can reduce forgetting as long as it aligns with the original data $D_A$ more than $D_B$. When $D_A$ and $D_B$ are already pointing in drastically opposite directions, making the term $\nabla\mathcal{L}(\theta_A; D_B)^T \nabla\mathcal{L}(\theta_A; D_A)$ negative, the mixing has a higher chance to lower $\Delta_2$. On the other hand, if $\nabla\mathcal{L}(\theta_A; D_B)^T \nabla\mathcal{L}(\theta_A; D_A)$ is positive, it is harder for mixing to mitigate forgetting.

# B SUPPLEMENTARY RESULTS

## B.1 MAIN RESULTS WITH STANDARD DEVIATION

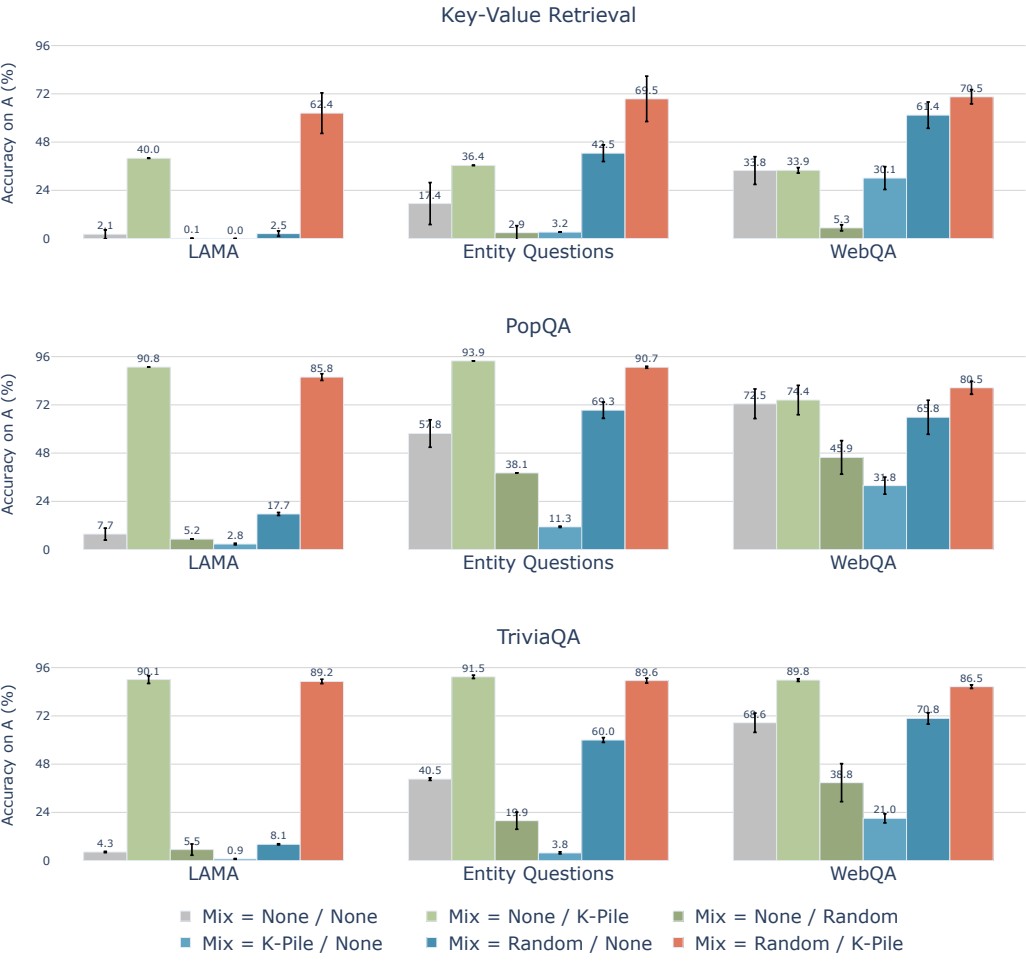

Figure 9: The main results of different combinations of $D_{M_1}/D_{M_2}$ over seed=[0,1,2] on the factoid datasets.

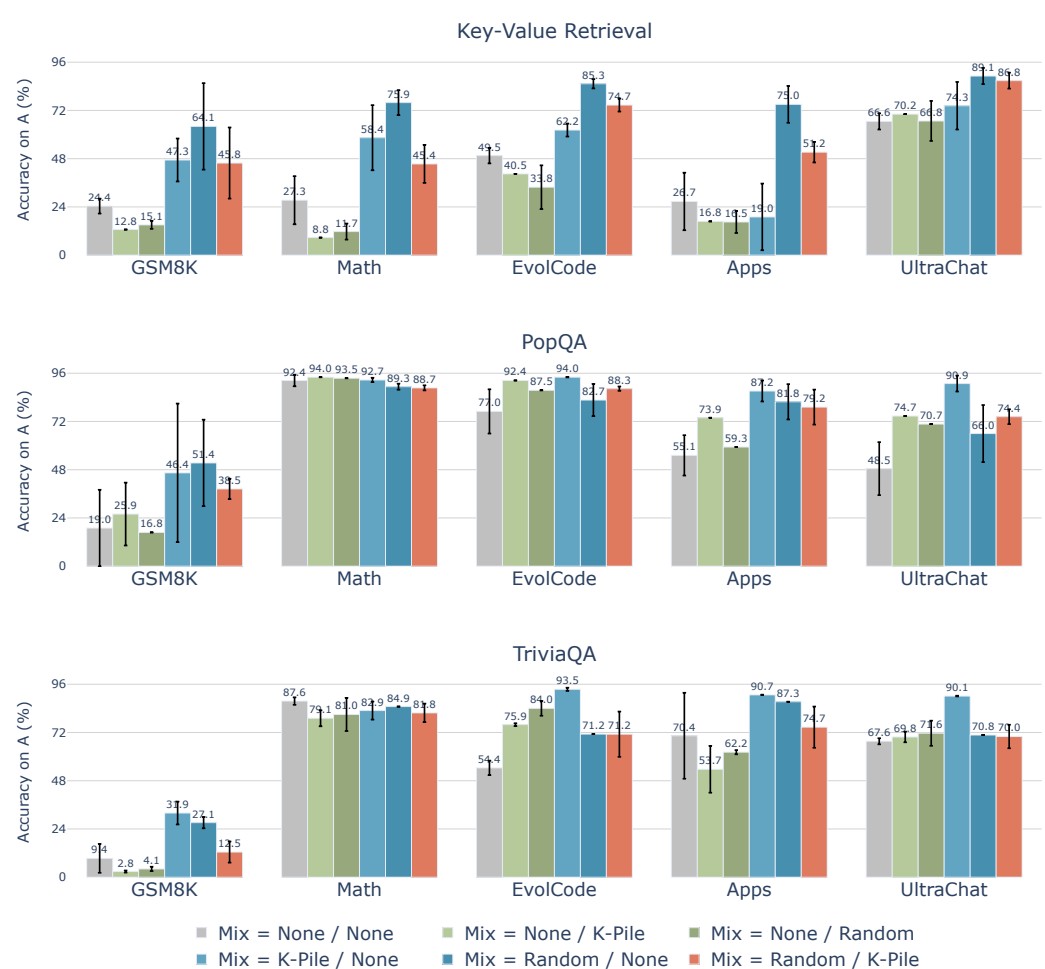

Figure 10: The main results of different combinations of $D_{M_1}/D_{M_2}$ over seed=[0,1,2] on the non-factoid datasets.

## B.2 REPLAY RESULTS

We report the full replay results in Table 4. Even though replay reduces more forgetting across the board, especially when we increase the ratio r, the replay-based method does not effectively mitigate forgetting in the factoid knowledge dataset.

## B.3 FORGETTING IN FAMILIAR FACTOID INSTANCES

We also investigate whether REMIX can retain the memorization of familiar factoid instances after directly fine-tuning on both factoid and non-factoid data in stage 2. After fine-tuning in stage 2, we evaluated the familiar instances from the factoid dataset $D_A$. The evaluation results for Llama-3-8B are shown in Table 5, and the results for Mistral-7B-v0.3 are presented in Table 3. We observe that mixing Knowledge-Pile, Arxiv-Pile, and FineWeb with factoid data in stage 2 helps mitigate the forgetting of familiar factoid instances for both Llama-3-8B and Mistral-7B-v0.3, aligning with the results in Figure 9.

## B.4 TRAINING DETAILS

In all experiments with Llama-3-8B, we average the results over three seeds [0, 1, 2], using a learning rate of 5e-5. For all experiments with Mistral-7B-v0.3, we use a learning rate of 1e-5. For the experiments on measure the forgetting of familiar factoid instances, we use a batch size of 128. For

| | LAMA | EntityQA | WebQA | GSM8K | Math | EvolCode | Apps | UltraChat |
|---|---|---|---|---|---|---|---|---|
| **Key-Value Recall** | | | | | | | | |
| Replay ($r = 0.00$) | **2.2** | **17.5** | **34.1** | **26.4** | **27.5** | **50.0** | **30.0** | **66.7** |
| Replay ($r = 0.01$) | 13.7 | 37.1 | 54.2 | 71.0 | 69.7 | 73.2 | 73.8 | 81.9 |
| Replay ($r = 0.05$) | 6.3 | 45.8 | 72.6 | 77.0 | 75.9 | 76.7 | 80.1 | 88.9 |
| Replay ($r = 0.1$) | 13.2 | 33.3 | 78.2 | 80.3 | 85.0 | 76.5 | 86.7 | 91.1 |
| **PopQA** | | | | | | | | |
| Replay ($r = 0.00$) | 15.7 | 64.3 | 78.6 | **33.6** | **93.5** | **80.5** | 63.2 | **53.7** |
| Replay ($r = 0.01$) | **12.0** | 66.0 | **75.3** | 94.4 | 95.1 | 95.7 | 90.8 | 87.6 |
| Replay ($r = 0.05$) | 27.4 | 64.4 | 84.5 | 95.9 | 95.2 | 95.4 | 95.9 | 95.3 |
| Replay ($r = 0.1$) | 46.6 | **64.0** | 83.8 | 96.1 | 96.0 | 95.7 | 96.3 | 95.7 |
| **TriviaQA** | | | | | | | | |
| Replay ($r = 0.00$) | 7.8 | 48.4 | 76.8 | **57.6** | 91.0 | **59.5** | 75.6 | **73.5** |
| Replay ($r = 0.01$) | **7.5** | **51.8** | 72.0 | 66.8 | 90.6 | 93.3 | **74.2** | 84.0 |
| Replay ($r = 0.05$) | 25.7 | 57.0 | 77.8 | 88.9 | 94.0 | 93.7 | 94.4 | 92.0 |
| Replay ($r = 0.1$) | 34.9 | 57.9 | 80.7 | 93.0 | 95.5 | 95.4 | 95.2 | 93.0 |

Table 4: The full Replay results at four replay ratio [0.0, 0.01, 0.05, 0.1].

| | LAMA | EntityQA | WebQA | GSM8K | Math | EvolCode | Apps | UltraChat |
|---|---|---|---|---|---|---|---|---|
| **PopQA** | | | | | | | | |
| / No Mixing | 27.3 | 24.4 | 39.1 | 13.0 | 18.3 | 36.3 | 7.4 | 46.9 |
| / K-Pile | 56.0 | 52.1 | 46.6 | 4.1 | 4.8 | 19.5 | 10.3 | 15.8 |
| / A-Pile | 65.1 | 60.4 | 52.7 | 9.2 | 3.2 | 26.5 | 21.8 | 19.3 |
| / Random | 24.9 | 27.9 | 29.1 | 7.5 | 6.0 | 25.4 | 2.8 | 18.4 |
| / FineWeb | 54.9 | 54.4 | 51.3 | 6.6 | 5.2 | 29.1 | 30.0 | 18.4 |
| **TriviaQA** | | | | | | | | |
| / No Mixing | 16.5 | 20.7 | 40.4 | 24.7 | 26.7 | 52.9 | 21.9 | 56.6 |
| / K-Pile | 55.9 | 57.5 | 50.3 | 11.0 | 6.8 | 28.4 | 20.9 | 23.4 |
| / A-Pile | 66.4 | 65.9 | 56.6 | 13.0 | 2.8 | 34.8 | 33.5 | 25.8 |
| / Random | 14.4 | 26.5 | 26.5 | 14.0 | 7.5 | 21.8 | 13.4 | 27.5 |
| / FineWeb | 56.6 | 57.6 | 52.6 | 13.0 | 6.0 | 38.9 | 56.9 | 17.7 |

Table 5: Llama-3-8B results for familiar-Factoid datasets. Mixing in stage 2 helps on maintaining the existing factoid instances.

| | LAMA | EntityQA | WebQA | GSM8K | Math | EvolCode | Apps | UltraChat |
|---|---|---|---|---|---|---|---|---|
| **PopQA** | | | | | | | | |
| / No Mixing | 55.5 | 47.1 | 68.0 | 14.5 | 42.8 | 25.9 | 18.4 | 38.9 |
| / K-Pile | 75.8 | 77.1 | 76.1 | 28.1 | 20.9 | 19.9 | 18.3 | 14.2 |
| / A-Pile | 78.2 | 79.0 | 77.6 | 28.1 | 20.9 | 19.9 | 18.3 | 14.2 |
| / Random | 52.7 | 53.8 | 62.6 | 28.1 | 20.9 | 19.9 | 18.3 | 14.2 |
| / FineWeb | 75.0 | 74.4 | 75.1 | 28.1 | 20.9 | 19.9 | 18.3 | 14.2 |
| **TriviaQA** | | | | | | | | |
| / No Mixing | 61.6 | 56.9 | 69.3 | 19.6 | 54.3 | 36.7 | 21.2 | 18.6 |
| / K-Pile | 79.1 | 79.6 | 73.9 | 29.3 | 20.8 | 20.6 | 20.7 | 18.6 |
| / A-Pile | 81.3 | 81.8 | 76.3 | 29.3 | 20.8 | 20.6 | 20.7 | 18.6 |
| / Random | 60.8 | 58.0 | 64.0 | 29.3 | 20.8 | 20.6 | 20.7 | 18.6 |
| / FineWeb | 78.8 | 80.0 | 73.7 | 29.3 | 20.8 | 20.6 | 20.7 | 18.6 |

Table 6: Mistral-7B-v0.3 results for familiar-Factoid datasets. Mixing in stage 2 helps on maintaining the existing factoid instances.

the rest of the experiments, we set the batch size to 32. Additionally, different stopping conditions are applied for the different factoid datasets: for the KVR task, we use a fixed number of epochs (20), while for other factoid tasks, training stops when the loss drops below 0.0001. We provide our training prompt in §B.5.

## B.5 INPUT-OUTPUT EXAMPLES FOR $D_A$ AND $D_M$

### B.5.1 INPUT-OUTPUT EXAMPLES FOR $D_A$

1. **Key-Value Recall ($D_A$):**
   Input text: The value of key *e6395973* is?
   Target text: *8219acf2*

2. **PopQA ($D_A$):**
   Input text: Question: What is New Lands's author? The answer is:
   Target text: Charles Fort

3. **TriviaQA ($D_A$):**
   Input text: Which city does David Soul come from? The answer is:
   Target text: Chicago

### B.5.2 INPUT-OUTPUT EXAMPLES FOR $D_M$

1. **Knowledge-Pile ($D_M$):**
   Input text:
   Complete the following partial passage: Processing hyperspectral images allows you to decode images and recognize objects in the scene on the base of analysis of spectrums. In some problems, information about the spectra may not be sufficient. In this case, visualization of data sets may use, for object recognition, by use additional non-formalized external attributes
   Target text:
   (for example, indicating the relative position of objects). Target visualization is a visualization adapted to a specific task of application. The method discussed in this chapter uses a way to visualize a measure of similarity to the sample. As a result of the transformation, the hyperspectral (multichannel) image is converted ...

2. **Arxiv-Pile ($D_M$):**
   Input text:
   Complete the following partial passage:
   — abstract: 'The purpose of this article is to study the problem of finding sharp lower bounds for the norm of the product of polynomials in the ultraproducts of Banach spaces $(X_i)_{\mathfrak{U}}$. We show that, under certain hypotheses, there is a strong relation between this problem and the same
   Target text:
   problem for the spaces $X_i$.' address: 'IMAS-CONICET' author: - Jorge Tomás Rodríguez title: On the norm of products of polynomials on ultraproducts of Banach spaces — Introduction =========== In this article we study the factor problem in the context of ultraproducts of Banach spaces. This problem can be stated as ...

3. **FineWeb ($D_M$):**
   Input text:
   Complete the following partial passage: *sigh* Fundamentalist community, let me pass on some advice to you I learned from the atheistic community: If you have set yourself on fire, do not run. Okay? Okay?? Please? Look, D, you had two months to say to Harvard in private emails, "I'm sorry, I shouldn't have been using
   Target text:
   that animation in my paid presentations. I wont use it again. I really do like 'Inner Life', though, and would love to use it in classroom presentations, from the BioVisions site, if that is acceptable." I sat here, for two months, waiting for that to happen, anything to happen, and ...

4. **Random Word Sequence ($D_M$):**
   Input text:

Memorize the following random-string passage:
pliosaur bismuth assertoric decentralization emerse redemonstrate sleepwaker Coracias thirstland Stercorariinae Cytherean autobolide pergamentaceous ophthalmodynamometer tensify tarefitch educement wime cockneity holotype spreng justiciary unseparate ascogonial chirimen Styphelia emotivity heller hystazarin unthinkable Corinth vicianose incommunicative sorcerous lineograph dochmiacal heresiographer interrenal anes mercal embryogenic swoon diptote funniness unwreathed contection rhapsodical infolding colorature multifurcate
Target text:
pliosaur bismuth assertoric decentralization emerse redemonstrate sleepwaker Coracias thirstland Stercorariinae Cytherean autobolide pergamentaceous ophthalmodynamometer tensify tarefitch educement wime cockneity holotype spreng justiciary unseparate ascogonial chirimen Styphelia emotivity heller hystazarin unthinkable Corinth vicianose incommunicative sorcerous lineograph dochmiacal heresiographer interrenal anes mercal embryogenic swoon diptote funniness unwreathed contection rhapsodical infolding colorature multifurcate

## B.6    DATASET DETAILS

We examine the strict overlap of knowledge entities between PopQA, TriviaQA, and the generic data used for mixing. By extracting knowledge entity pairs from the questions and target answers, we calculate the exact overlap between these pairs. The overlap percentage among PopQA, TriviaQA, and the generic data is less than 1.3%.

## B.7    PROBING RESULTS

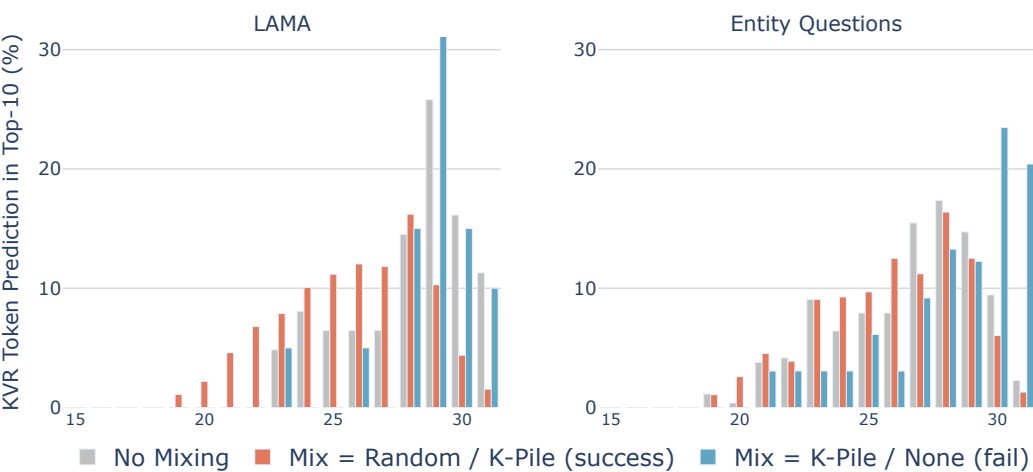

Figure 11: Probing of the Key-Value Recall task. x-axis: layer index. y-axis: the normalized frequency of the correct token occurring in the top-10 tokens probed at each layer.

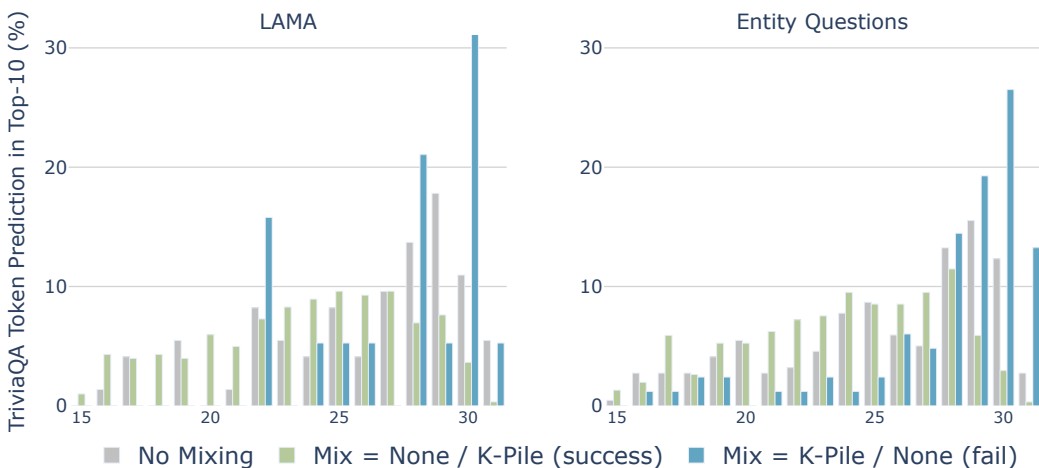

Figure 12: Probing of the TriviaQA task. x-axis: layer index. y-axis: the normalized frequency of the correct token occurring in the top-10 tokens probed at each layer.

