# OpenReview forum: "Continual Memorization of Factoids in Large Language Models"
_ICLR.cc/2025/Conference — Submitted to ICLR 2025_

### Official Review · Reviewer_Qd9Z · 2024-10-31

**Soundness:** 2
**Presentation:** 3
**Contribution:** 2
**Rating:** 5
**Confidence:** 4

**Summary:**

This paper examines the problem of forgetting in large language models (LLMs) during continual learning, particularly when training on a small set of long-tail factoids (subject-relation-object triples). The authors identify two primary challenges in retaining these long-tail facts over successive training stages: the limitations of standard replay techniques and the interference from training on unrelated datasets. To address these challenges, the authors propose REMIX (Random and Generic Data Mixing), which combines unrelated, generic data with the factoid data to prevent forgetting. Through comprehensive experiments, REMIX is shown to outperform replay-based methods and recover performance from severe forgetting. The authors further analyze how REMIX influences the learning process, noting that it shifts the storage of factoids to earlier layers and diversifies the layers used for storing these facts, thus reducing interference from later training stages.

**Strengths:**

Novel Approach to Memory Retention: REMIX introduces a unique approach to mitigate forgetting by mixing random and generic data during training, achieving substantial performance improvement compared to replay-based methods.
Thorough Experimental Analysis: The authors conduct extensive experiments across multiple datasets, providing empirical evidence of REMIX’s effectiveness. They also analyze layer-specific behavior, offering insights into how REMIX modifies the model’s memory dynamics.
Generalizable Insight for Continual Learning: By demonstrating the limitations of replay techniques and proposing alternative strategies, this paper offers valuable insights for both continual memory retention and general continual learning in LLMs.

**Weaknesses:**

1 Lack of Comparison with Other Forgetting Mitigation Techniques:
Although the authors discuss the limitations of replay-based methods, the paper lacks a systematic comparison with other common forgetting mitigation techniques, such as Elastic Weight Consolidation (EWC) or Knowledge Distillation. For instance, EWC is frequently used in continual learning to reduce interference by regularizing key weights, while Knowledge Distillation selectively retains critical information. Comparing REMIX with these methods would help clarify REMIX’s unique advantages and performance under similar conditions.
2 Synthetic and Specific Dataset Selection:
The datasets used in this paper, such as Key-Value Recall and PopQA, are primarily synthetic and consist of isolated factoids, which may not fully reflect the complexity of real-world data. For example, in practical scenarios, knowledge is often presented in overlapping or nested forms (e.g., “The author of Hamlet is Shakespeare” and “Shakespeare wrote Hamlet”) rather than as isolated facts. Testing REMIX on more commonly used datasets, such as Wikipedia or open-domain QA datasets (e.g., Natural Questions), could provide a more realistic evaluation of its effectiveness and generalizability.
3 Unclear Justification for Types of Data Mixing:
The paper employs both random word sequences and knowledge-rich text (e.g., Knowledge Pile) as mixed data to prevent forgetting, but it does not provide a clear explanation of why these two disparate types would produce similar effects. For example, random word sequences contain no factual content, while Knowledge Pile includes a substantial amount of knowledge and contextual information. The authors could further analyze why both random and knowledge-rich data help prevent forgetting or test the specific impacts of each type in different scenarios.
4 Impact on Performance in New Tasks:
While REMIX performs well in retaining early-stage knowledge, the paper does not explore its impact on subsequent new tasks. For instance, it would be useful to know whether REMIX might limit the model's ability to learn these new tasks when introduced for fine-tuning. Evaluating REMIX’s impact on new tasks could provide insights into potential trade-offs between memory retention and generalization to new tasks.
5 Limited Evaluation on Extended Stages:
The experiments primarily focus on two-stage continual learning, with limited testing of multi-stage scenarios. In real-world applications, models may undergo multiple updates, such as continual fine-tuning in legal or medical domains. Testing REMIX in a three-stage or four-stage setting could provide better insight into its stability and effectiveness over longer training cycles.
6 Resource and Scalability Concerns:
REMIX relies on incorporating additional mixed data during training, which may increase computational costs, especially for large models such as Llama-3-8B. Expanding this method to resource-intensive domains like finance or healthcare could present challenges. If the authors could discuss the trade-offs between added data usage and computational demands or provide a rough estimate of the resources required to implement REMIX in a real-world setting, it would help assess its feasibility and scalability in high-resource environments.

**Questions:**

1 Comparison with Other Forgetting Mitigation Techniques:
How does REMIX compare with other established forgetting mitigation methods across different tasks? A systematic comparison would strengthen the case for REMIX’s advantages.
2 Exploration of Bidirectional Relational Memory with Atomic Fact Datasets:
The datasets used appear to consist mainly of isolated factoids or "atomic" facts, without directly exploring bidirectional or inverse relational memory. For example, if the model learns that "Bob’s dad is Sam," it would be valuable to evaluate whether the model can infer the inverse relationship, such as "Who is Sam's son?" This type of associative memory is essential for comprehensive fact retention, as it reflects a more integrated understanding of relationships. Could the authors clarify whether such tests were conducted, or suggest if REMIX could potentially extend to this type of bidirectional memorization?
3 Why Forgetting is More Pronounced with Factoid Datasets:
The paper reports that models experience significant forgetting when fine-tuned on factoid datasets in the second stage, but not on non-factoid datasets. Could the authors elaborate on why forgetting is more pronounced with factoids compared to non-factoids, as well as any observed differences in how REMIX performs on these types? This could provide further insight into the underlying mechanisms of forgetting and the strengths of REMIX.
4 Rationale Behind Data Mixing Types:
The paper employs various data sources (e.g., Knowledge Pile, random word sequences) as mixed data in REMIX. However, the choice of these sources appears empirical, lacking theoretical justification or detailed explanation. It remains unclear why certain data sources yield better performance on specific tasks, and this potential variation across tasks is not fully explored. There is no clear guideline for selecting mixed data types, nor an analysis of how different types of mixed data impact task performance. A more thorough theoretical or empirical examination of these differences could enhance understanding of REMIX’s applicability and effectiveness across various contexts.
5 Impact of REMIX on New Task Performance:
The paper focuses on preventing forgetting in prior tasks, but it does not discuss the potential impact of REMIX on performance for new tasks introduced in later stages. While REMIX seems effective at preserving knowledge from earlier stages, it remains unclear whether this approach might inadvertently reduce performance on new tasks due to constraints placed on the model’s capacity or flexibility. An analysis of how REMIX affects the model's performance on new tasks would provide a more balanced understanding of its effectiveness in continual learning contexts.
6 Effectiveness of Random vs. Generic Text Mixing:
The paper explores both random word sequence mixing and generic pretraining text mixing in REMIX. However, it is not entirely clear whether these two approaches yield similar or differing effects on knowledge retention. Could the authors provide more details on any observed differences in effectiveness between random and generic data mixing? Understanding how each type impacts forgetting could offer valuable insights into the dynamics of memory retention in large language models.
7 Combined Mixing Effectiveness:
The results indicate that combining random word sequence mixing with generic data mixing produces the best outcomes, but it is not fully explained why this combination is most effective. Is there a theoretical or empirical rationale for why mixing both types of data provides better retention compared to using either one alone? Additional explanation of this combined effect would enhance understanding of REMIX’s underlying mechanisms and may help guide future applications.
8 100% Accuracy in Table 1:
In Table 1, it is stated that all Stage 1 datasets are trained to 100% accuracy before Stage 2 training. Could the authors clarify how this 100% accuracy is achieved and guaranteed across different datasets? Specifically, were there particular training techniques or criteria used to ensure full memorization of Stage 1 data? Additional details on this process would help in understanding the baseline setup for evaluating forgetting.
9 Suitability Across Task Types:
Has REMIX been tested on other types of tasks, such as generative or dialogue-based tasks? Additional testing on these tasks would clarify REMIX’s versatility and applicability beyond factoid retention.

---

> ### Author Response · Authors · 2024-11-18
> **Response to the reviewer (1/4)**
>
> We thank the reviewer for the detailed comments, suggestions, and the recognition of the novelty of our methods, the extensiveness of our experiments, and the generalizability of our conclusions. The reviewer pointed out several important aspects that would help strengthen our investigation, which we provide further experiments and explanations in the following passages.
>
> &nbsp;
> > [...] the paper lacks a systematic comparison with other common forgetting mitigation techniques, such as Elastic Weight Consolidation (EWC) or Knowledge Distillation.
>
> We provide three more types of baselines to compare with our data mixing method:
> - Weight regularization: we use Elastic Weight Consolidation (EWC) [1] and calculate the Fisher score using one backward pass using the current mini-batch for training.
> - Behavior regularization: we add the KL between the training model vs the original reference model to the loss. Knowledge distillation can be seen as a type of regularization in the continual learning setting [12].
> - Parameter expansion method: we learn separate and none-overlapping LoRA adapters at stage 1 and 2, similar to the IncLoRA model in [13].
> We compare these baselines against the No Mixing baseline and REMIX (Random at stage 1 and Knowledge Pile at stage 2). We show results on the datasets that *suffer most from forgetting*: all factoid datasets and GSM8K from the non-factoid datasets.
>
>
> | KVR                     | LAMA | EntityQA | WebQA | GSM8K | Avg  |
> | ----------------------- | ---- | -------- | ----- | ----- | ---- |
> | No Mixing               | 2.1  | 17.4     | 33.8  | 22.4  | 18.9 |
> | REMIX (Random/KP)       | 62.4 | 69.5     | 70.2  | 45.8  | **62.0** |
> | Weight Regularization   | 0.1  | 4.3      | 76.7  | 2.6   | 20.9 |
> | Behavior Regularization | 0.2  | 15.6     | 36.6  | 28.1  | 20.1 |
> | Parameter Expansion     | 0.0  | 0.0      | 0.0   | 0.0   | 0.0  |
>
> | PopQA                   | LAMA | EntityQA | WebQA | GSM8K | Avg  |
> | ----------------------- | ---- | -------- | ----- | ----- | ---- |
> | No Mixing               | 7.7  | 57.8     | 72.5  | 19.0  | 39.3 |
> | REMIX (Random/KP)       | 85.8 | 90.7     | 80.5  | 38.5  | **73.9** |
> | Weight Regularization   | 12.1 | 67.4     | 76.7  | 25.7  | 45.5 |
> | Behavior Regularization | 7.5  | 59.3     | 55.5  | 40.6  | 40.7 |
> | Parameter Expansion     | 0.0  | 0.1      | 0.0   | 1.2   | 0.3  |
>
> | TriviaQA                | LAMA | EntityQA | WebQA | GSM8K | Avg  |
> | ----------------------- | ---- | -------- | ----- | ----- | ---- |
> | No Mixing               | 4.3  | 40.5     | 68.6  | 9.4   | 30.7 |
> | REMIX (Random/KP)       | 89.2 | 89.6     | 86.5  | 12.5  | **69.5** |
> | Weight Regularization   | 7.9  | 58.5     | 80.3  | 37.9  | 46.2 |
> | Behavior Regularization | 6.8  | 39.0     | 71.0  | 14.5  | 32.8 |
> | Parameter Expansion     | 21.9 | 0.1      | 1.1   | 3.0   | 6.5  |
>
>
> We observe that the weight regularization baseline and output regularization baseline can obtain better factoid retention at different tasks but on average lags behind REMIX by a large margin (40%+ on KVR, 30%+ on PopQA, and 20%+ on TriviaQA). In our attempt, the parameter expansion based baseline learns to achieve 100% accuracy at stage 2, but catastrophically forgets at stage 2, achieving close to zero factoid retention.

---

> ### Author Response · Authors · 2024-11-18
> **Response to the reviewer (2/4)**
>
> > The datasets used in this paper, such as Key-Value Recall and PopQA, are primarily synthetic and consist of isolated factoids, which may not fully reflect the complexity of real-world data. [...] Testing REMIX on more commonly used datasets, such as Wikipedia or open-domain QA datasets (e.g., Natural Questions), could provide a more realistic evaluation of its effectiveness and generalizability.
>
> We totally agree that capturing real-world complexities is important to understand the generalizability of our conclusion. In our investigation, we aim to strike a balance between controllability and the real world complexity. With the Key-Value Recall (KVR) task, we can ensure there’s no data contamination from pretraining despite its simplicity. On the other hand, both PopQA and TriviaQA are sourced from real websites yet maintain the “atomic” nature of each factoid. While PopQA is still template-based, questions in TriviaQA is sourced from real Trivia websites and annotated by humans, which is close to how Natural Question is collected. We aimed to understand the memorization dynamics of these cases from simple/controllable (KVR) to realistic (TriviaQA).
>
> Nonetheless, we fully recognize the importance of using more natural and realistic datasets to strengthen our conclusion suggested by the reviewer. We further investigate the effectiveness of REMIX on the Natural Question dataset as suggested. The results are the following:
>
>
> | Natural Questions  | LAMA | EntityQA | WebQA | GSM8K | MATH | EvolCode | APPS | UltraChat | Avg  |
> | ------------------ | ---- | -------- | ----- | ----- | ---- | -------- | ---- | --------- | ---- |
> | No Mixing          | 16.2 | 80.2     | 90.0  | 71.8  | 88.3 | 99.2     | 93.9 | 87.6      | 78.4 |
> | REMIX (Random/-)  | 38.5 | 81.2     | 63.6  | 80.1  | 90.9 | 79.9     | 89.3 | 80.8      | 75.5 |
> | REMIX (KP/-)      | 2.8  | 13.8     | 55.6  | 93.5  | 99.2 | 99.2     | 99.8 | 98.5      | 70.3 |
> | REMIX (-/Random)  | 18.7 | 61.7     | 85.6  | 78.3  | 96.1 | 92.7     | 96.1 | 88.5      | 77.2 |
> | REMIX (-/KP)      | 94.4 | 95.7     | 91.4  | 75.7  | 94.8 | 93.9     | 96.8 | 83.9      | **90.8** |
> | REMIX (Random/KP) | 94.1 | 95.7     | 91.7  | 56.7  | 93.1 | 86.6     | 78.2 | 84.6      | 85.1 |
>
> KP = Knowledge Pile.
> The trend aligns with the trends for our choice of datasets in the paper – mixing Knowledge Pile at stage 2 leads to best retention of the learned factoids (+12.4% average accuracy over No Mixing), followed by mixing Random Word Sequence at stage 1 + Knowledge Pile at stage 2 (+6.7% average accuracy over No Mixing). We observe that forgetting is generally less severe than the datasets we chose, which might be due to some level of contamination reported in the Llama 3 technical report (Table 15).
>
> &nbsp;
> > Unclear Justification for Types of Data Mixing: The paper employs both random word sequences and knowledge-rich text (e.g., Knowledge Pile) as mixed data to prevent forgetting, but it does not provide a clear explanation of why these two disparate types would produce similar effects. [...] However, the choice of these sources appears empirical, lacking theoretical justification or detailed explanation. It remains unclear why certain data sources yield better performance on specific tasks, and this potential variation across tasks is not fully explored.
>
> We motivate our choice of different mixing strategies in section 4.1 (L260 - L267) with the following intuition: REMIX at stage 1 aims to *protect* the learned factoids by maintaining a good starting weight space for stage 2 training. REMIX at stage 2 aims to *reduce the interference* of the memorized factoids.
> With this intuition, we justify the use of the two mixing datasets with mathematical derivation in Appendix A.3 (L866 - L917). Specifically, in stage 1, we need to choose a mixing data that is uncorrelated to $D_A$ and $D_B$, hence the choice of random word sequence. In stage 2, any natural distribution can achieve mitigation when forgetting is already severe; when the dataset aligns with $D_A$ more than $D_B$, the mitigation can be more effective (e.g., Knowledge Pile helps knowledge factoids more than Arxiv Pile in Figure 5).
>
> &nbsp;
> > Impact on Performance in New Tasks: While REMIX performs well in retaining early-stage knowledge, the paper does not explore its impact on subsequent new tasks. For instance, it would be useful to know whether REMIX might limit the model's ability to learn these new tasks when introduced for fine-tuning.
>
> When applying mixing at stage 2, we ensure convergence for dataset $D_B$ using the same stopping criterion (loss reaching below 0.0001 by 5 times). We did not observe issues lowering the loss after stage 1 training. Similarly, we observe no issue when performing stage 3 training with mixing data as well (L426-L431 and Figure 8). This suggests that mixing data does not hinder the model's ability to learn new tasks.

---

> > ### Comment · Reviewer_Qd9Z · 2024-12-02
> >
> > I appreciate the authors' recognition of the need for more realistic datasets, and the additional experiments on Natural Questions are valuable. The results for Natural Questions with REMIX are indeed promising, and I can see how they align with the trends observed in the original datasets. That said, I am still curious about how REMIX might perform on datasets that involve overlapping or nested knowledge (e.g., relationships between facts or entities that require reasoning beyond isolated factoid recall). It would be interesting to explore whether REMIX can handle more complex knowledge integration in such contexts.

---

> ### Author Response · Authors · 2024-11-18
> **Response to the reviewer (3/4)**
>
> > Expanding this method to resource-intensive domains like finance or healthcare could present challenges. If the authors could discuss the trade-offs between added data usage and computational demands [...], it would help assess its feasibility and scalability in high-resource environments.
>
> Scalability is indeed critical for practical adaptations as the more mixing data needed for mitigation, the less scalable and practical it is. We provide analysis on the amount of REMIX data needed for successful forgetting mitigation. We show in L421-L424 the mixing ratio required for different mixing scenarios: most REMIX strategies only need a mixing ratio of 1.0 to be effective, and increasing the mixing ratio hits diminishing returns.
> We also simulate the scaled-up scenario (resource intensive cases as suggested) by increasing the dataset size from 2000 factoids to 4000 factoids.
>
> | PopQA with n=4000        | LAMA | EntityQA | WebQA | GSM8K |
> | ------------------------ | ---- | -------- | ----- | ----- |
> | No Mixing                | 9.2  | 69.0     | 69.1  | 61.2  |
> | REMIX ratio=1.0 (-/KP)  | 89.8 | 93.1     | 81.0  | 65.2  |
> | REMIX ratio=2.0 (-/KP)  | 89.3 | 93.4     | 80.8  | 40.9  |
> | REMIX ratio=4.0 (-/KP)  | 88.9 | 92.2     | 77.0  | 55.5  |
>
> KP = Knowledge Pile.
> We find that the same trend holds for the scaled up case where only the mixing ratio of 1.0 is effective. A promising future direction is to discover the kind of mixing data that can be effective with very small mixing ratios.
>
> &nbsp;
> > Exploration of Bidirectional Relational Memory with Atomic Fact Datasets: [...] Could the authors clarify whether such tests were conducted, or suggest if REMIX could potentially extend to this type of bidirectional memorization?
>
> The reviewer rightfully pointed out the importance of transferring the learned knowledge to downstream tasks to make it useful. This is discussed in depth in works such as [2, 3, 4, 5], which highlight the difficulty of manipulating the learned knowledge in downstream tasks. Our work is mainly positioned as the *prerequisite* before knowledge manipulation – if the knowledge is not retained in the first place, then there’s no chance for it to be recalled and manipulated successfully. While retention does not entail successful manipulation, we aim to understand the dynamics of retention as the first step, which is on its own a challenging task.
>
> Nonetheless, we fully recognize the importance of this question and further conducted evaluations to assess such capabilities of our models. We use three templates to assess the model’s ability to manipulate learned knowledge on the KVR task (since it is guaranteed to have no contamination from pretraining):
>
> Template 1 (reverse recall): \
> The key of the value DEF is? \
> Key1: ABC, Value: DEF \
> Answer: ABC
>
> Template 2 (selective recall): \
> Here are two keys: ABC and XYZ. What is the value of the first key? \
> Key1: ABC, Value1: DEF \
> Key2: XYZ, Value2: GHI \
> Answer: DEF
>
> Template 3 (recall-then-manipulate): \
> If the first character in the value of key ABC is changed to X, what is the new value of key? \
> Key: ABC \
> Value: DEF \
> Answer: XEF
>
> We evaluate on the following models: No-Mixing, REMIX (Random / -), and REMIX (Random / Knowledge-Pile).
>
> | KVR        |               | LAMA | EntityQA | WebQA | GSM8K | MATH | EvolCode | APPS | UltraChat | Avg  |
> | ---------- | ------------- | ---- | -------- | ----- | ----- | ---- | -------- | ---- | --------- | ---- |
> | Template 1 | No Mixing     | 0.0  | 0.0      | 0.0   | 0.0   | 0.0  | 0.0      | 0.0  | 0.0       | 0.0  |
> |            | REMIX (R/-)  | 0.0  | 0.0      | 0.0   | 0.0   | 0.0  | 0.0      | 0.0  | 0.0       | 0.0  |
> |            | REMIX (R/KP) | 0.0  | 0.0      | 0.0   | 0.0   | 0.0  | 0.0      | 0.0  | 0.0       | 0.0  |
> | Template 2 | No Mixing     | 0.3  | 1.6      | 0.3   | 2.9   | 7.3  | 1.3      | 15.8 | 15.8      | 5.7  |
> |            | REMIX (R/-)  | 0.0  | 1.6      | 4.2   | 9.4   | 3.5  | 70.9     | 26.6 | 35.2      | 18.9 |
> |            | REMIX (R/KP) | 7.1  | 18.2     | 8.4   | 1.9   | 0.0  | 0.3      | 5.3  | 36.6      | 9.7  |
> | Template 3 | No Mixing     | 0.0  | 0.1      | 0.6   | 0.3   | 0.5  | 0.0      | 0.6  | 5.1       | 0.9  |
> |            | REMIX (R/-)  | 0.0  | 0.8      | 1.7   | 1.2   | 1.2  | 3.4      | 0.8  | 5.0       | 1.8  |
> |            | REMIX (R/KP) | 3.1  | 4.0      | 3.1   | 0.4   | 0.0  | 0.0      | 2.6  | 6.1       | 2.4  |
>
> R = Random Word Sequence. KP = Knowledge Pile.
> We observe that none of the models can perform Template 1, which corroborates with [8, 9, 10, 11], highlighting the unique challenge of reverse recall in knowledge storage and manipulation.
> However, we found that REMIX improves other types of knowledge manipulation such as Template 2 and 3 as shown in the table. This is an interesting finding that warrants further study. We appreciate the reviewer’s suggestion and will include this and further findings in the updated version of the paper.

---

> > ### Comment · Reviewer_Qd9Z · 2024-12-02
> >
> > Concerning the choice of data mixing types, I now understand the motivation for using both random word sequences and knowledge-rich text like Knowledge Pile. The mathematical derivation and intuition provided in the rebuttal are helpful. However, I still have a few questions about the empirical impact of these two types of mixing. For example, does the performance differ significantly when using only one of these mixing types (e.g., Random vs. Knowledge Pile) across various tasks, and under what specific conditions might one be more effective than the other?

---

> ### Author Response · Authors · 2024-11-18
> **Response to the reviewer (4/4)**
>
> > Testing REMIX in a three-stage or four-stage setting could provide better insight into its stability and effectiveness over longer training cycles.
>
> We investigated the three-stage setting (L426 - L431 and Figure 8) and show that REMIX can retain learned factoids for different combinations of stage 2 data.
>
> &nbsp;
> > Why forgetting is more pronounced with factoids compared to non-factoids, as well as any observed differences in how REMIX performs on these types?
>
> While we do not have a direct mechanistic explanation for this phenomenon, it corroborates with the findings in 1) the continual learning literature which suggest catastrophic forgetting happens when two tasks are similar and therefore interfere [6, 7], and 2) finetuning on unfamiliar knowledge disrupts the model and causes exacerbated hallucinations [8, 9, 10, 11]. A mechanistic understanding of this phenomenon is an important area for future investigation but is slightly outside of the scope of our paper.
>
> &nbsp;
> > Effectiveness of Random vs. Generic Text Mixing: The paper explores both random word sequence mixing and generic pretraining text mixing in REMIX. However, it is not entirely clear whether these two approaches yield similar or differing effects on knowledge retention. Could the authors provide more details on any observed differences in effectiveness between random and generic data mixing?
>
> Both the Random and Generic mixing strategies can be applied at either stage 1 or stage 2 and they yield different results as shown in Table 2. Specifically, mixing Random is only effective in stage 1 and mixing Generic is only effective in stage 2. When combining both it yields the best results. This aligns with our motivation in section 4.1 – mixing uncorrelated data in stage 1 *protects* the learned factoids and mixing at stage 2 helps *reduce interference* especially when forgetting is severe and the mixing data aligns with $D_A$.
>
> &nbsp;
> > 100% Accuracy in Table 1: In Table 1, it is stated that all Stage 1 datasets are trained to 100% accuracy before Stage 2 training. Could the authors clarify how this 100% accuracy is achieved and guaranteed across different datasets? Specifically, were there particular training techniques or criteria used to ensure full memorization of Stage 1 data?
>
> We train the models until reaching a loss below 0.0001 by 5 times before stopping to ensure full convergence. This often entails training many epochs to guarantee perfect accuracy. We provide training details in Appendix B.4 for reference.
>
> &nbsp;
> > Has REMIX been tested on other types of tasks, such as generative or dialogue-based tasks?
>
> In our initial exploration, we experimented with generative tasks and found that the forgetting phenomenon is much less pronounced than long-tail factoid data to begin with. This prompted us to further investigate the problem of forgetting specific to factoid datasets. We see the same phenomenon manifested in stage 2 training as well where non-factoid datasets generally have less impact leading to forgetting.
>
> &nbsp;
> &nbsp;
>
> Please let us know if there's any more information we can provide to clarify our work, thank you!
>
> References
>
> [1] Kirkpatrick et al., Overcoming Catastrophic Forgetting in Neural Networks. PNAS 2017.
>
> [2] Yang et al., Synthetic Continued Pretraining. Arxiv 2024.
>
> [3] Allen-Zhu and Li, Physics of Language Models: Part 3.1, Knowledge Storage and Extraction. Arxiv 2024.
>
> [4] Allen-Zhu and Li, Physics of Language Models: Part 3.2, Knowledge Manipulation. Arxiv 2024.
>
> [5] Berglund et al., The Reversal Curse: LLMs trained on "A is B" fail to learn "B is A". ICLR 2024.
>
> [6] Farajtabar et al., Orthogonal gradient descent for continual learning. AISTATS 2020.
>
> [7] Bennani et al., Generalisation Guarantees for Continual Learning with Orthogonal Gradient Descent. ICML 2020.
>
> [8] Kang et al., Unfamiliar Finetuning Examples Control How Language Models Hallucinate. Arxiv 2024.
>
> [9] Gekhman et al., Does Fine-Tuning LLMs on New Knowledge Encourage Hallucinations? EMNLP 2024.
>
> [10] Zhang et al., Knowledge Overshadowing Causes Amalgamated Hallucination in Large Language Models. Arxiv 2024.
>
> [11] Ghosal et al., Understanding Finetuning for Factual Knowledge Extraction. ICLR 2024.
>
> [12] Sun et al., Distill and Replay for Continual Language Learning. COLING 2020.
>
> [13] Wang et al., Orthogonal Subspace Learning for Language Model Continual Learning. EMNLP 2023.

---

> > ### Author Response · Authors · 2024-11-22
> >
> > Dear Reviewer,
> >
> > We'd like to send a gentle reminder that we have submitted the rebuttal to address your comments. We sincerely appreciate your feedback and are happy to address any additional questions you may have during this discussion period.
> >
> > We thank you again for taking the time to review our work.

---

> > ### Comment · Reviewer_Qd9Z · 2024-12-02
> >
> > In terms of scalability and resource concerns, I appreciate the authors' efforts to evaluate the impact of increasing data sizes and mixing ratios. It would be helpful to know if there are any guidelines or recommendations regarding the trade-offs between computational cost and the effectiveness of REMIX, especially in domains where resources are constrained. For instance, how does REMIX perform when applied to larger models or when deployed in environments with limited computational resources?

---

> > > ### Author Response · Authors · 2024-12-04
> > > **Response to Reviewer Qd9Z (1/2)**
> > >
> > > > I appreciate the authors' recognition of the need for more realistic datasets, and the additional experiments on Natural Questions are valuable. The results for Natural Questions with REMIX are indeed promising, and I can see how they align with the trends observed in the original datasets. That said, I am still curious about how REMIX might perform on datasets that involve overlapping or nested knowledge (e.g., relationships between facts or entities that require reasoning beyond isolated factoid recall). It would be interesting to explore whether REMIX can handle more complex knowledge integration in such contexts.
> > >
> > >
> > >
> > > Thank you for the acknowledgement that the experiment on Natural Questions as the more realistic dataset shows to be promising and valuable.
> > >
> > > In the experiments provided in the previous comment, we show that three tests where different templates are used to evaluate REMIX’s ability to manipulate stored knowledge beyond simple recall. For example, reverse recall (template 1) is the “Bob’s dad is Sam” vs “Sam’s son is Bob” case. We show that REMIX performs better than No Mixing at *selective recall (template 2)* and *recall-then-manipulate (template 3)*, suggesting that REMIX can already improve on some forms of knowledge manipulation even if it is only trained on isolated examples.
> > > REMIX still fails at reverse recall (template 1), which highlights that some forms of knowledge manipulation are more challenging.
> > >
> > > While we would like to emphasize that the major challenge we aim to address in this work is knowledge retention, we are happy to include more evaluation on knowledge manipulation in the updated paper if the reviewer has further suggestions on any particular dataset.
> > >
> > > > Concerning the choice of data mixing types, I now understand the motivation for using both random word sequences and knowledge-rich text like Knowledge Pile. The mathematical derivation and intuition provided in the rebuttal are helpful. However, I still have a few questions about the empirical impact of these two types of mixing. For example, does the performance differ significantly when using only one of these mixing types (e.g., Random vs. Knowledge Pile) across various tasks, and under what specific conditions might one be more effective than the other?
> > >
> > > Yes, mixing with random word sequence (Random) vs Generic (e.g., K-Pile) data differs empirically as we show this in Table 2 in the paper: Random is most effective when mixing at stage 1, and K-Pile is most effective when mixed in stage 2.
> > >
> > > Take Key Value Recall for example, mixing Random at stage 1 improves the factoid accuracy ($13.5 \rightarrow 28.8$), but mixing Random at stage 2 does not help ($13.5  \rightarrow 2.1$). Conversely, mixing K-Pile at stage 1 does not help ($13.5  \rightarrow 8.4$), but mixing K-Pile at stage 2 improves significantly ($13.5  \rightarrow 27.8$). This trend is consistent when training on *factoid* datasets in stage 2.
> > >
> > > When training on *non-factoid* datasets at stage 2, Random is still best to mix at stage 1 ($38.9  \rightarrow 77.9$) as opposed to mixing at stage 2 ($38.9  \rightarrow 28.8$), and K-Pile is best to mix at stage 1 as well ($38.9  \rightarrow 52.2$) as opposed to stage 2 ($38.9  \rightarrow 29.8$).
> > >
> > > In short, mixing Random at stage 1 is most effective, but mixing K-Pile at stage 1 does not help for factoid data and should be mixed in stage 2, while for non-factoid data K-Pile is more effective at stage 1. This aligns with the intuition suggested by our derivation.

---

> > > > ### Author Response · Authors · 2024-12-04
> > > > **Response to Reviewer Qd9Z (2/2)**
> > > >
> > > > We can show the effectiveness of REMIX under constrained computational resources in the following two cases: 1) effectiveness of REMIX on smaller models, or 2) reducing computational needs by lowering the mixing ratio when the model size or data size is large.
> > > >
> > > > For the first case, we provide REMIX experiments on TriviaQA with different model sizes below.
> > > >
> > > > | Llama-3.2-1B        | LAMA | EntityQA | WebQA | GSM8K | MATH | EvolCode | APPS | UltraChat | Avg  |
> > > > | ----------------------- | ---- | -------- | ----- | ----- | ---- | -------- | ---- | --------- | ---- |
> > > > | No Mixing               | 36.7 | 79.5     | 91.9  | 88.9  | 97.0 | 98.8     | 98.8 | 97.9      | 86.2 |
> > > > | Mixing ratio=1.0 (-/KP) | 97.6 | 96.7     | 96.4  | 94.9  | 97.8 | 98.9     | 98.8 | 97.9      | **97.4** |
> > > > | Mixing ratio=4.0 (-/KP) | 98.2 | 98.2     | 95.5  | 97.2  | 97.6 | 97.4     | 98.5 | 95.3      | 97.2 |
> > > >
> > > >
> > > > | Llama-3.2-3B            | LAMA | EntityQA | WebQA | GSM8K | MATH | EvolCode | APPS | UltraChat | Avg  |
> > > > | ----------------------- | ---- | -------- | ----- | ----- | ---- | -------- | ---- | --------- | ---- |
> > > > | No Mixing               | 47.4 | 79.5     | 91.9  | 88.9  | 93.4 | 98.8     | 98.3 | 97.6      | 87.0 |
> > > > | Mixing ratio=1.0 (-/KP) | 95.9 | 95.1     | 94.7  | 94.1  | 95.9 | 98.8     | 98.8 | 97.3      | 96.3 |
> > > > | Mixing ratio=4.0 (-/KP) | 98.3 | 98.3     | 96.7  | 96.4  | 98.1 | 98.0     | 96.6 | 95.8      | **97.3** |
> > > >
> > > > We observe that REMIX is very effective across mixing ratios. Under constrained resources where only small models can be served, REMIX maintains its effectiveness.
> > > >
> > > > For the second case, we showed that 1) the mixing ratio need not scale with dataset size in our previous response and 2) a mixed ratio of 1.0 is effective across different model sizes, from 1B to 3B to 8B in our paper, indicating that the ratio does not increase as model size scales.
> > > >
> > > > We also aim to provide results on larger model sizes, which we are unable to provide at this moment due to time and resource limits. We will include the larger model experiment in our updated version of our paper.

---

> > > > > ### Author Response · Authors · 2024-12-04
> > > > >
> > > > > We would like to thank the reviewer again for the time for reviewing our work, providing actionable feedback, and engaging in the discussion. We hope our responses address your concerns and would appreciate if the scores can be updated to reflect it.

---

### Official Review · Reviewer_hvCG · 2024-11-02

**Soundness:** 3
**Presentation:** 3
**Contribution:** 2
**Rating:** 3
**Confidence:** 4

**Summary:**

The paper tackles the issue of continual memorization of factoids in large language models (LLMs), focusing on retaining specific, rare knowledge (factoids) as the model undergoes further training on unrelated datasets. Typical replay techniques fail to prevent forgetting of such factoids in LLMs, leading the authors to propose REMIX, a data-mixing approach that interleaves random or generic data during training stages to reduce forgetting. The paper demonstrates that REMIX helps preserve factoid knowledge across various datasets and training scenarios, with results analyzed using tools like Logit Lens.

**Strengths:**

Originality: The paper draws attention to the issue of continual memorization of long-tail information through factoid memorization.
Quality: The experiments are conducted rigorously, covering a range of datasets and demonstrating REMIX’s impact across several configurations.
Clarity: Explanations are mostly clear, and the figures help illustrate key points.
Significance: The method has some practical relevance for fact retention.

**Weaknesses:**

Unclear Problem Motivation: The paper does not convincingly explain why memorizing long-tail knowledge in the form of factoids is important in practical applications. Without a clear motivation, the relevance of the problem formulation is uncertain, which diminishes the contribution’s significance. If we are only concerned about factoids, why use LLMs in the first place? Why not just use traditional knowledge-based systems? The authors should show how memorizing factoids leads to downstream applications, such as utilizing the information from the factoids on tasks that specifically require LLMs.
Lack of Novelty: REMIX lacks sufficient originality; the idea of mixing generic data into training is not groundbreaking and does not specifically address the unique challenges of factoid memorization.
Lack of Baselines: The authors only explore experience replay as the baseline approaches, whereas there exists other methods in literature that can mitigate forgetting during continued pretraining (parameter expansion-based methods, regularization methods, etc.)

**Questions:**

Suggestions
- I would like to suggest the authors the strengthen the motivation of needing to memorize long-tail knowledge through the form of factoids, but showing that it transfers the knowledge itself to downstream NLP tasks that require integrating those long-tail information. Simply getting a high score in the factoid task itself is insufficient to motivate the problem formulation.
- I would suggest that the authors include more baselines from the continual learning literature that can mitigate the forgetting of previously learned knowledge.

---

> ### Author Response · Authors · 2024-11-18
> **Response to the reviewer (1/3)**
>
> We thank the reviewer for recognizing the rigor in our experimentation and the clearness in our writing. We also appreciate that the reviewer pointed out several aspects that would benefit from further clarification and experimentation to solidify the arguments in our investigation. We provide the explanations and further evidence in the following passages.
>
> &nbsp;
> > The paper does not convincingly explain why memorizing long-tail knowledge in the form of factoids is important in practical applications. [...] If we are only concerned about factoids, why use LLMs in the first place? Why not just use traditional knowledge-based systems? The authors should show how memorizing factoids leads to downstream applications, such as utilizing the information from the factoids on tasks that specifically require LLMs.
>
> We agree that the choice between parametric vs non-parametric representation for factoid knowledge has a long standing tension in practical scenarios. However, we position our work more as part of the larger body of works that aim to understand the knowledge acquisition dynamics of language models through finetuning and the unintended risk ([1, 2, 3, 4]), which also evaluate on knowledge datasets such as PopQA and TriviaQA.
>
> The reviewer rightfully pointed out the importance of transferring the learned knowledge to downstream tasks to make it useful. This is discussed in depth in works such as [5, 6, 7, 8] which highlight the difficulty of manipulating the learned knowledge in downstream tasks. Our work is positioned as the *prerequisite* before knowledge manipulation – if the knowledge is not retained in the first place, then there’s no chance for it to be recalled and manipulated successfully. While retention does not entail successful manipulation, we aim to understand the dynamics of retention as the first step, which is on its own a challenging task.
> We appreciate this profound point and will improve our framing to reflect this emphasis and make the distinction clearer.
>
> Nonetheless, we fully recognize the importance of this question and further conducted evaluations to assess such capabilities of our models. We use three templates to assess the model’s ability to manipulate learned knowledge on the KVR task (since it is guaranteed to have no contamination from pretraining):
>
> Template 1 (reverse recall): \
> The key of the value DEF is? \
> Key1: ABC, Value: DEF \
> Answer: ABC
>
> Template 2 (selective recall): \
> Here are two keys: ABC and XYZ. What is the value of the first key? \
> Key1: ABC, Value1: DEF \
> Key2: XYZ, Value2: GHI \
> Answer: DEF
>
> Template 3 (recall-then-manipulate): \
> If the first character in the value of key ABC is changed to X, what is the new value of key? \
> Key: ABC \
> Value: DEF \
> Answer: XEF
>
> We evaluate on the following models: No-Mixing, REMIX (Random / -), and REMIX (Random / Knowledge-Pile).
>
> | KVR        |               | LAMA | EntityQA | WebQA | GSM8K | MATH | EvolCode | APPS | UltraChat | Avg  |
> | ---------- | ------------- | ---- | -------- | ----- | ----- | ---- | -------- | ---- | --------- | ---- |
> | Template 1 | No Mixing     | 0.0  | 0.0      | 0.0   | 0.0   | 0.0  | 0.0      | 0.0  | 0.0       | 0.0  |
> |            | REMIX (R/-)  | 0.0  | 0.0      | 0.0   | 0.0   | 0.0  | 0.0      | 0.0  | 0.0       | 0.0  |
> |            | REMIX (R/KP) | 0.0  | 0.0      | 0.0   | 0.0   | 0.0  | 0.0      | 0.0  | 0.0       | 0.0  |
> | Template 2 | No Mixing     | 0.3  | 1.6      | 0.3   | 2.9   | 7.3  | 1.3      | 15.8 | 15.8      | 5.7  |
> |            | REMIX (R/-)  | 0.0  | 1.6      | 4.2   | 9.4   | 3.5  | 70.9     | 26.6 | 35.2      | 18.9 |
> |            | REMIX (R/KP) | 7.1  | 18.2     | 8.4   | 1.9   | 0.0  | 0.3      | 5.3  | 36.6      | 9.7  |
> | Template 3 | No Mixing     | 0.0  | 0.1      | 0.6   | 0.3   | 0.5  | 0.0      | 0.6  | 5.1       | 0.9  |
> |            | REMIX (R/-)  | 0.0  | 0.8      | 1.7   | 1.2   | 1.2  | 3.4      | 0.8  | 5.0       | 1.8  |
> |            | REMIX (R/KP) | 3.1  | 4.0      | 3.1   | 0.4   | 0.0  | 0.0      | 2.6  | 6.1       | 2.4  |
>
> R = Random Word Sequence. KP = Knowledge Pile.
> We observe that none of the models can perform Template 1, which corroborates with [7, 8], highlighting the unique challenge of reverse recall in knowledge storage and manipulation.
> However, we found that REMIX improves other types of knowledge manipulation such as Template 2 and 3 as shown in the table. This is an interesting finding that warrants further study. We appreciate the reviewer’s suggestion and will include this and further findings in the updated version of the paper.

---

> ### Author Response · Authors · 2024-11-18
> **Response to the reviewer (2/3)**
>
> > the idea of mixing generic data into training is not groundbreaking and does not specifically address the unique challenges of factoid memorization
>
> We would like to first point out the important distinctions between our setting and the general continual learning setting where mixing is applied. 1) Mixing often assumes access to distributions of the previous training stages (similar to Replay in Section 3.2 that uses a small percentage of $D_A$). In REMIX, the model does not have access to $D_A$, which renders the problem much more challenging. 2) Mixing random sequence data is unexplored in past literature and its effectiveness is surprising. 3) While mixing generic pretraining data is familiar in settings like continual pretraining, its effectiveness is largely under-explored in the context of factual knowledge retention. 4) Although it appears fairly straightforward, we discovered that such a simple strategy can work extremely well, which we established through mathematical derivation and validated through extensive experimentation.
>
> Along with the empirical efficacy of REMIX, we provide intuition (section 4.1) and derivations (Appendix A.3) to justify the choice of random data (help protecting the learned knowledge) and the generic data (help reduce the interference of the stage 2 data), which we hope supplement the existing data mixing methods such as replay.
>
> &nbsp;
> > The authors only explore experience replay as the baseline approaches, whereas there exists other methods in literature that can mitigate forgetting during continued pretraining (parameter expansion-based methods, regularization methods, etc.
>
> We provide three more types of baselines to compare with our data mixing method:
> - Weight regularization: we use Elastic Weight Consolidation (EWC) and calculate the Fisher score using one backward pass using the current mini-batch for training [9].
> - Behavior regularization: we add the KL between the training model vs the original reference model to the loss [10].
> - Parameter expansion method: we learn separate and none-overlapping LoRA adapters at stage 1 and 2, similar to the IncLoRA model in [11].
>
> We compare these baselines against the No Mixing baseline and REMIX (Random at stage 1 and Knowledge Pile at stage 2). We show results on the datasets that *suffer most from forgetting*: all factoid datasets and GSM8K from the non-factoid datasets.
>
>
> | KVR                     | LAMA | EntityQA | WebQA | GSM8K | Avg  |
> | ----------------------- | ---- | -------- | ----- | ----- | ---- |
> | No Mixing               | 2.1  | 17.4     | 33.8  | 22.4  | 18.9 |
> | REMIX (Random/KP)       | 62.4 | 69.5     | 70.2  | 45.8  | **62.0** |
> | Weight Regularization   | 0.1  | 4.3      | 76.7  | 2.6   | 20.9 |
> | Behavior Regularization | 0.2  | 15.6     | 36.6  | 28.1  | 20.1 |
> | Parameter Expansion     | 0.0  | 0.0      | 0.0   | 0.0   | 0.0  |
>
> | PopQA                   | LAMA | EntityQA | WebQA | GSM8K | Avg  |
> | ----------------------- | ---- | -------- | ----- | ----- | ---- |
> | No Mixing               | 7.7  | 57.8     | 72.5  | 19.0  | 39.3 |
> | REMIX (Random/KP)       | 85.8 | 90.7     | 80.5  | 38.5  | **73.9** |
> | Weight Regularization   | 12.1 | 67.4     | 76.7  | 25.7  | 45.5 |
> | Behavior Regularization | 7.5  | 59.3     | 55.5  | 40.6  | 40.7 |
> | Parameter Expansion     | 0.0  | 0.1      | 0.0   | 1.2   | 0.3  |
>
> | TriviaQA                | LAMA | EntityQA | WebQA | GSM8K | Avg  |
> | ----------------------- | ---- | -------- | ----- | ----- | ---- |
> | No Mixing               | 4.3  | 40.5     | 68.6  | 9.4   | 30.7 |
> | REMIX (Random/KP)       | 89.2 | 89.6     | 86.5  | 12.5  | **69.5** |
> | Weight Regularization   | 7.9  | 58.5     | 80.3  | 37.9  | 46.2 |
> | Behavior Regularization | 6.8  | 39.0     | 71.0  | 14.5  | 32.8 |
> | Parameter Expansion     | 21.9 | 0.1      | 1.1   | 3.0   | 6.5  |
>
> We observe that the weight regularization baseline and output regularization baseline can obtain better factoid retention at different tasks but on average lags behind REMIX by a large margin (40%+ on KVR, 30%+ on PopQA, and 20%+ on TriviaQA). In our attempt, the parameter expansion based baseline learns to achieve 100% accuracy at stage 2, but catastrophically forgets at stage 2, achieving close to zero factoid retention.

---

> ### Author Response · Authors · 2024-11-18
> **Response to the reviewer (3/3)**
>
> Please let us know if there's any more information we can provide to clarify our work, thank you!
>
>
> References
>
> [1] Kang et al., Unfamiliar Finetuning Examples Control How Language Models Hallucinate. Arxiv 2024.
>
> [2] Gekhman et al., Does Fine-Tuning LLMs on New Knowledge Encourage Hallucinations? EMNLP 2024.
>
> [3] Zhang et al., Knowledge Overshadowing Causes Amalgamated Hallucination in Large Language Models. Arxiv 2024.
>
> [4] Ghosal et al., Understanding Finetuning for Factual Knowledge Extraction. ICLR 2024.
>
> [5] Yang et al., Synthetic Continued Pretraining. Arxiv 2024.
>
> [6] Allen-Zhu and Li, Physics of Language Models: Part 3.1, Knowledge Storage and Extraction. Arxiv 2024.
>
> [7] Allen-Zhu and Li, Physics of Language Models: Part 3.2, Knowledge Manipulation. Arxiv 2024.
>
> [8] Berglund et al., The Reversal Curse: LLMs trained on "A is B" fail to learn "B is A". ICLR 2024.
>
> [9] Kirkpatrick et al., Overcoming Catastrophic Forgetting in Neural Networks. PNAS 2017.
>
> [10] Sun et al., Distill and Replay for Continual Language Learning. COLING 2020.
>
> [11] Wang et al., Orthogonal Subspace Learning for Language Model Continual Learning. EMNLP 2023.

---

> > ### Author Response · Authors · 2024-11-22
> >
> > Dear Reviewer,
> >
> > We'd like to send a gentle reminder that we have submitted the rebuttal to address your comments. We sincerely appreciate your feedback and are happy to address any additional questions you may have during this discussion period.
> >
> > We thank you again for taking the time to review our work.

---

### Official Review · Reviewer_XLf9 · 2024-11-03

**Soundness:** 4
**Presentation:** 4
**Contribution:** 3
**Rating:** 8
**Confidence:** 4

**Summary:**

This work focuses on the continual memorization setting in LLMs, a subsetting of continual learning in which a model is first fine-tuned on a factoid dataset, then successively fine-tuned on other datasets (multiple training stages) and must retain knowledge learned during the first stage. The authors first demonstrate that catastrophic forgetting occurs in a 2-stage training process, especially if the dataset from the second stage is a factoid one, and that usual replay methods used in continual learning do not satisfactorily mitigate the issue.

The authors then introduce REMIX, a strategy for preventing forgetting in the multi-stage learning process. In this strategy, additional training data is added to one or both of the training stages. This data takes the form of either generic of random data. The authors show that this new method produces significantly better result than the basic training process on LLaMa-3 and Mistral-7B, which they show to be linked to a more diversified and larger set of layers in which factoids are stored.

Finally, the authors perform a large number of validation experiments, proving that this method is effective with different mixing datasets, and investigate the effect of several other hyperparameters such as the mixing sequence length, the mixing ratio and the number of training stages.

**Strengths:**

- The paper is very well-written and clear. The section order feels natural, and the reasoning and intuition for each idea are given clearly. The tables and charts summarize the data well, and while informative, the text flows well and is not overly complicated. As a result, this is a very pleasant paper to read. In addition, the references are recent and seem relevant.

- The paper touches upon the issue of catastrophic forgetting of factoids in LLMs, which is a relevant and unsolved issue, especially in the current context where many pre-trained LLMs showcase good reasoning capabilities but cannot easily be updated afterward to store novel world knowledge.

- The paper contains a large number of experiments, that give clear motivation for introducing REMIX, and then show its efficacy over many settings.

- The ideas found in this work are not revolutionary per se (which is not to say that they lack originality; see my next point), but the execution is straightforward and good. The authors carefully checked for important details such as dataset overlap.

- The idea of mixing generic/random data with the training dataset is quite creative and original. Despite being counterintuitive, the authors justify this idea mathematically.

As a result, I recommend this paper for publication with no major point of criticism.

**Weaknesses:**

Many of the points of criticism I had while reading this paper were answered later on, or in the appendices. The other points that I have mainly consist of questions (see section below).

- In section 4.2, the word "Figure" is used several times instead of "Table".
- Section 3.2 (on replay) is lacking detail in comparison to other sections, especially as it justifies the use of REMIX compared to other replay methods. In particular, I could not find which of the two LLMs was used to measure the effect of replay methods.

**Questions:**

- The authors show that their method causes the model to store factoids in more layers of the model, which presumably means that the factoids overwrite previous data in these shallower layers. It would have been interesting to investigate whether this results in any significant degradation of other model capabilities (e.g. fluency) compared to the basic two-stage training process. I understand that this paper specifically focuses on factoid memorization and contains many experiments already, but this could be mentioned as future work.

- Another interesting experiment would be to vary either the model or the dataset's size, to evaluate the link between model capacity and the efficacy of REMIX/replay techniques. Do the authors have any insight or early intuition regarding this?

- Have the authors considered/tried combining REMIX with classic replay techniques? This seems like a natural next step to know whether the use of both methods leads to even better results.

---

> ### Author Response · Authors · 2024-11-18
> **Response to the reviewer**
>
> We thank the reviewer for the positive comments, especially recognition of the importance of the problem, the novelty of our proposed mitigation method, and the extensiveness of our experiments. We also appreciate the reviewer for the constructive and actionable suggestions.
>
> &nbsp;
> > Section 3.2 (on replay) is lacking detail in comparison to other sections, especially as it justifies the use of REMIX compared to other replay methods. In particular, I could not find which of the two LLMs was used to measure the effect of replay methods.
>
> Thank you for pointing this out. The model used for replay was the Llama 3 8B model. We did not directly compare replay and REMIX due to the difference in their settings – replay assumes access to stage 1 dataset $D_A$ and REMIX does not. Therefore, we use replay mainly as inspiration for REMIX as opposed to a competitive baseline. For clarity as suggested, we will update the REMIX section to refer and draw appropriate comparison to the replay method.
>
> &nbsp;
> > It would have been interesting to investigate whether this results in any significant degradation of other model capabilities (e.g. fluency) compared to the basic two-stage training process. I understand that this paper specifically focuses on factoid memorization and contains many experiments already, but this could be mentioned as future work.
>
> We find models trained with REMIX actually contain slightly better fluency compared to other models mainly because the models without mixing suffer from overfitting more severely. However, we totally agree a more comprehensive analysis on the side effects when using REMIX will be very valuable. We will aim to include such an analysis in the updated version.
>
> &nbsp;
> > [...] vary either the model or the dataset's size to evaluate the link between model capacity and the efficacy of REMIX/replay techniques.
>
> We provide an extra result on increasing the $D_A$ dataset size from 2000 factoids to 4000 factoids ($D_B$ size maintains to be 2000).
>
> | PopQA with n=4000        | LAMA | EntityQA | WebQA | GSM8K |
> | ------------------------ | ---- | -------- | ----- | ----- |
> | No Mixing                | 9.2  | 69.0     | 69.1  | 61.2  |
> | REMIX ratio=1.0 (-/KP)  | 89.8 | 93.1     | 81.0  | 65.2  |
> | REMIX ratio=2.0 (-/KP)  | 89.3 | 93.4     | 80.8  | 40.9  |
> | REMIX ratio=4.0 (-/KP)  | 88.9 | 92.2     | 77.0  | 55.5  |
>
> KP = Knowledge Pile.
> We find that the effectiveness of REMIX holds for the scaled up dataset size case. We also find that there seems to be generally less forgetting in the No Mixing case, which might be an interesting phenomenon to further study.
> With the computation budgets at hand we defer the model scaling experiments to future research.
>
> &nbsp;
> > Have the authors considered/tried combining REMIX with classic replay techniques? This seems like a natural next step to know whether the use of both methods leads to even better results.
>
> In our investigation, we aim to fully separate the two settings: 1) allowing access to $D_A$ at later stages, and 2) strict continual learning setting where $D_A$ cannot be used in stage 2. We use the second setting throughout since using $D_A$ can be seen as a form of *cheating* because they are the exact factoids to memorize. Therefore, Replay (section 3.2) was only meant to motivate REMIX instead of treated as a fully comparable baseline.
>
> Please let us know if there's any more information we can provide to clarify our work, thank you!

---

> > ### Comment · Reviewer_XLf9 · 2024-11-21
> >
> > I thank the authors for their clarifications and their extensive responses to the rebuttals, as well as for the additional experimental results they have provided. I have read in detail the other reviewers' comments and rebuttals, and maintain my current assessment of the authors' work. Furthermore, I strongly encourage the authors to update their manuscript to include their additional experiments (e.g. as appendices).

---

> > > ### Author Response · Authors · 2024-12-04
> > >
> > > We thank the reviewer for the acknowledgement. We will update the manuscript to include their additional experiments as suggested.

---

### Official Review · Reviewer_QUxn · 2024-11-04

**Soundness:** 3
**Presentation:** 3
**Contribution:** 3
**Rating:** 5
**Confidence:** 4

**Summary:**

This paper studied the forgetting issues when finetuning a LLM in multi-stage datasets. They focus the setting of continual memorizing of factoid facts - Stage 1 is factoid fact datasets and Stage 2 finetune with fact/non-fact datasets. The authors find non-fact datasets will cause smaller drop. Based on this intuition, the authors proposed a data mixing strategy (introducing some unrelated datasets) in multi-stage fine-tuning to reduce the forgetting.

**Strengths:**

The papers present problems and solution pretty clear and easy to follows.
The authors proposed a simple yet effect way to reduce the interference among the different fine-tuning datasets.

**Weaknesses:**

There are some other method to reduce the interference among the datasets of stage 1 and stage 2.  For example, the method needs to compare with another baseline, i.e. "mixing of Data A and Data B"

**Questions:**

Table 1, the degradation is more severe for Stage 2 is also a factoid dataset. Do you have any explanation? Also, there is big drop when using GSK8k. It will be very insightful to understand the interplays of the datasets.

For the Replay approach, what if we use a ratio = 1.0?

---

> ### Author Response · Authors · 2024-11-18
> **Response to the reviewer (1/2)**
>
> We thank the reviewer for the comments and suggestions. We appreciate your positive comments on the clarity and simplicity of our problem setup and the proposed solutions.
>
>
> > There are some other methods to reduce the interference among the datasets of stage 1 and stage 2. For example, the method needs to compare with another baseline, i.e. "mixing of Data A and Data B"
>
> Thank you for pointing out the need for more baselines in order to strengthen our conclusions.
>
> We would like to clarify that since we focus on understanding how the knowledge of D_A is retained after another stage of training,  stage 2 shouldn't have access to D_A (including mixing D_A and D_B).
>
>
> Nonetheless, we recognize that more baselines can strengthen our conclusions. We provide three more types of representative baselines to compare with our data mixing method:
> - Weight regularization: we use Elastic Weight Consolidation (EWC) and calculate the Fisher score using one backward pass using the current mini-batch for training [1].
> - Behavior regularization: we add the KL between the training model vs the original reference model to the loss [8].
> - Parameter expansion method: we learn separate and none-overlapping LoRA adapters at stage 1 and 2, similar to the IncLoRA model in [9].
>
> We compare these baselines against the No Mixing baseline and REMIX (Random at stage 1 and Knowledge Pile at stage 2). We show results on the datasets that *suffer most from forgetting*: all factoid datasets and GSM8K from the non-factoid datasets.
>
>
> | KVR                     | LAMA | EntityQA | WebQA | GSM8K | Avg  |
> | ----------------------- | ---- | -------- | ----- | ----- | ---- |
> | No Mixing               | 2.1  | 17.4     | 33.8  | 22.4  | 18.9 |
> | REMIX (Random/KP)       | 62.4 | 69.5     | 70.2  | 45.8  | **62.0** |
> | Weight Regularization   | 0.1  | 4.3      | 76.7  | 2.6   | 20.9 |
> | Behavior Regularization | 0.2  | 15.6     | 36.6  | 28.1  | 20.1 |
> | Parameter Expansion     | 0.0  | 0.0      | 0.0   | 0.0   | 0.0  |
>
> | PopQA                   | LAMA | EntityQA | WebQA | GSM8K | Avg  |
> | ----------------------- | ---- | -------- | ----- | ----- | ---- |
> | No Mixing               | 7.7  | 57.8     | 72.5  | 19.0  | 39.3 |
> | REMIX (Random/KP)       | 85.8 | 90.7     | 80.5  | 38.5  | **73.9** |
> | Weight Regularization   | 12.1 | 67.4     | 76.7  | 25.7  | 45.5 |
> | Behavior Regularization | 7.5  | 59.3     | 55.5  | 40.6  | 40.7 |
> | Parameter Expansion     | 0.0  | 0.1      | 0.0   | 1.2   | 0.3  |
>
> | TriviaQA                | LAMA | EntityQA | WebQA | GSM8K | Avg  |
> | ----------------------- | ---- | -------- | ----- | ----- | ---- |
> | No Mixing               | 4.3  | 40.5     | 68.6  | 9.4   | 30.7 |
> | REMIX (Random/KP)       | 89.2 | 89.6     | 86.5  | 12.5  | **69.5** |
> | Weight Regularization   | 7.9  | 58.5     | 80.3  | 37.9  | 46.2 |
> | Behavior Regularization | 6.8  | 39.0     | 71.0  | 14.5  | 32.8 |
> | Parameter Expansion     | 21.9 | 0.1      | 1.1   | 3.0   | 6.5  |
>
>
>
> We observe that the weight regularization baseline and output regularization baseline can obtain better factoid retention at different tasks but on average lags behind REMIX by a large margin (40%+ on KVR, 30%+ on PopQA, and 20%+ on TriviaQA). In our attempt, the parameter expansion based baseline learns to achieve 100% accuracy at stage 2, but catastrophically forgets at stage 2, achieving close to zero factoid retention.
>
>
> > For the Replay approach, what if we use a ratio = 1.0?
>
> Similar to the previous point, a ratio of 1.0 means using the entire $D_A$ together with $D_B$ for training at stage 2, which is in conflict with the continual multi-stage setting. Replay is not meant to be a baseline it uses D_A but rather as a motivational study that inspires REMIX. In the factoid memorization task, using $D_A$ is essentially *cheating*, so it should be understood as a different settings (therefore 10% reported in section 3.2 seems adequate for the motivational purpose).
>
> > Table 1, the degradation is more severe for Stage 2 is also a factoid dataset. Do you have any explanation? Also, there is a big drop when using GSK8k. It will be very insightful to understand the interplays of the datasets.
>
> While it is hard to obtain a direct mechanistic explanation for this phenomenon, it corroborates with the findings in 1) the continual learning literature which suggest catastrophic forgetting happens when two tasks are similar and therefore interfere [2, 3], and 2) finetuning on unfamiliar knowledge disrupts the model and causes exacerbated hallucinations [4, 5, 6, 7]. A mechanistic understanding of this phenomenon is an important area for future investigation but is slightly outside of the scope of our paper.
>
> For GSM8K, we hypothesize that the special format of its training data (e.g., extensive use of angle brackets) might contribute to forgetting as the model picks up such irregularities very quickly.

---

> ### Author Response · Authors · 2024-11-18
> **Response to the reviewer (2/2)**
>
> Please let us know if there's any more information we can provide to clarify our work, thank you!
>
> Reference:
>
> [1] Kirkpatrick et al., Overcoming Catastrophic Forgetting in Neural Networks. PANS 2017.
>
> [2] Farajtabar et al., Orthogonal gradient descent for continual learning. AISTATS 2020.
>
> [3] Bennani et al., Generalisation Guarantees for Continual Learning with Orthogonal Gradient Descent. ICML 2020.
>
> [4] Kang et al., Unfamiliar Finetuning Examples Control How Language Models Hallucinate. Arxiv 2024.
>
> [5] Gekhman et al., Does Fine-Tuning LLMs on New Knowledge Encourage Hallucinations? EMNLP 2024.
>
> [6] Zhang et al., Knowledge Overshadowing Causes Amalgamated Hallucination in Large Language Models. Arxiv 2024.
>
> [7] Ghosal et al., Understanding Finetuning for Factual Knowledge Extraction. ICLR 2024.
>
> [8] Sun et al., Distill and Replay for Continual Language Learning. COLING 2020.
>
> [9] Wang et al., Orthogonal Subspace Learning for Language Model Continual Learning. EMNLP 2023.

---

> ### Author Response · Authors · 2024-11-22
>
> Dear Reviewer,
>
> We'd like to send a gentle reminder that we have submitted the rebuttal to address your comments. We sincerely appreciate your feedback and are happy to address any additional questions you may have during this discussion period.
>
> We thank you again for taking the time to review our work.

---

### Author Response · Authors · 2024-11-22
**General Message to Reviewers**

Dear Reviewers,

With the author-reviewer discussion period ending soon, we wanted to take this opportunity to thank you again for your suggestions and to check whether you had any remaining questions after our responses! We especially hope that you have a chance to review the additional experiments we ran based on reviewer suggestions described in each of our rebuttals.

If we've addressed your concerns, we'd be grateful if you'd consider updating your score!

---

### Meta-Review · Area_Chair_UxB4 · 2024-12-24

**Metareview:**

Summary: The paper investigates the challenges of continual memorization in large language models (LLMs), focusing on retaining long-tail factual knowledge (factoids) across multiple stages of training. It highlights that standard fine-tuning approaches often lead to catastrophic forgetting of these factoids, especially when subsequent training involves similar datasets. The authors propose a mitigation strategy, REMIX (Random and Generic Data Mixing), which mixes random or unrelated generic data into training stages to preserve factoid knowledge. REMIX outperforms traditional replay methods, enhancing retention by altering where and how factoids are stored within the model.

Strengths:
- Clear writing and presentation
- Simplicity of the approach

Weakness:
- Lukewarm response from all but one reviewer and the positive reviewer didn't champion the paper
- Primarily focused on isolated factoids rather than more complex knowledge relationships
- More comprehensive analysis on the side effects when using REMIX is not presented
- Limited exploration of performance on downstream tasks and more realistic datasets
- Applicability of the results with scaling is not clear based on further experiments provided during rebuttal as forgetting seems to reduce
- Resource requirements and scalability considerations could be better addressed

Decision: Given the lack of enthusiasm from the reviewers and limited practical relevance, unfortunately, the paper can't be accepted in its current form and addressing all the concerns would warrant another round of reviewing.

**Additional Comments On Reviewer Discussion:**

We thank the authors and reviewers for engaging during the discussion phase towards improving the paper. Below are some of the highlights:

1. Multiple reviewers (including the positive one) want to see more ablations and baselines:
- Reviewers asked for simpler ablations and baselines like mixing and more replay settings, e.g. with ratio=1.
- Authors instead added comparisons with weight regularization, behavior regularization, and parameter expansion methods.

2. Side-effects of REMIX
- Reviewers asked if REMIX affects other capabilities of the model like fluency
- Authors provided a qualitative response mentioning fluency improves and left a more comprehensive analysis on the side effects when using REMIX for future work

3. Questions on practical relevance
- Reviewers asked if limited to factoid why not use traditional knowledge systems
- Authors argued the work belonged to the larger body of works that aim to understand the knowledge acquisition dynamics of language models through finetuning
- Reviewers also asked for more datasets like NQ,
- Authors conducted additional experiments on Natural Questions dataset
- Finally reviewers asked if factoids can be retrieved in reverse order (bidirectional) and other knowledge manipulations
- Authors instead provided experiments via templates, showing improvements in selective recall but not in reverse recall

4. Scalability concerns: Authors provided results on smaller models and more data. The results were mixed as even though REMIX remains effective the gap or forgetting itself seems to reduce.

5. Theoretical understanding: Authors expanded on the mathematical intuition and empirically validated when different mixing strategies work best.

---

### Decision · Program_Chairs · 2025-01-22

Reject